# Iterative Sketching and its Application to Federated Learning

## Abstract

Johnson-Lindenstrauss lemma is one of the most valuable tools in machine learning, since it enables the reduction to the dimension of various learning problems. In this paper, we exploit the power of Fast-JL transform or so-called sketching technique and apply it to federated learning settings. Federated learning is an emerging learning scheme which allows multiple clients to train models without data exchange. Though most federated learning frameworks only require clients and the server to send gradient information over the network, they still face the challenges of communication efficiency and data privacy. We show that by iteratively applying independent sketches combined with additive noises, one can achieve the above two goals simultaneously. In our designed framework, each client only passes a sketched gradient to the server, and de-sketches the average-gradient information received from the server to synchronize. Such framework enjoys several benefits: 1). Better privacy, since we only exchange randomly sketched gradients with low-dimensional noises, which is more robust against emerging gradient attacks; 2). Lower communication cost per round, since our framework only communicates low-dimensional sketched gradients, which is particularly valuable in a small-bandwidth channel; 3). No extra overall communication cost. We provably show that the introduced randomness does not increase the overall communication at all.

## 1 Introduction

Federated learning enables multiple parties to collaboratively train a machine learning model without directly exchanging training data. This has become particularly important in areas of artificial intelligence where users care data privacy, security, and access rights, including healthcare (Li et al., 2020b; 2019), internet of things (Chen et al., 2020), and fraud detection (Zheng et al., 2020).

Given the importance and popularity of federated learning, it has become an important research topic for academia and industry, mostly focusing on two central themes. One is on data privacy. Federated learning seemingly protects clients' privacy, since it only communicates gradient information. Unfortunately, recent studies (Geiping et al., 2020; Zhu & Han, 2020; Wang et al., 2019) have demonstrated that attackers can recover the input data from the shared gradients. The reason why the attacks work is the gradients carry important information about the training data (Ateniese et al., 2015; Fredrikson et al., 2015). The second one is on communication efficiency. Machine learning models are becoming increasingly larger but client devices that carry private data only have limited network bandwidth. The size of the gradient is the same as the size of the model, and the amount of data to communication between the clients and the servers thus is large. This becomes even more problematic when conducting federated learning on mobile and edge devices, where the bandwidth of the network is low. The communication cost is one of the most important the key performance bottlenecks in federated learning systems Goga & Teixeira (2012); Konečný et al. (2016). Many works try to address this challenge through local optimization method, such as local gradient descent (GD), local stochastic gradient descent (SGD) and their variants (Konečný et al., 2016; McMahan et al., 2017; Stich, 2018).

Despite existing efforts, no work addresses both challenges simultaneously as far as we concern. Therefore, we ask the following question:

*Is there a FL framework that protects the local privacy and tackles the communication challenge?*

In this paper, we achieve these goals by using an old but powerful idea — the Johnson-Lindenstrauss transform and its fast variations (Fast-JL, see Ailon & Chazelle (2006)). If we view the transform as a sketch matrix, and its transpose as de-sketch, then our main idea is to iteratively apply the sketch and de-sketch matrices to gradients. Instead of running the vanilla gradient descent $w^{(t+1)} \leftarrow w^{(t)} - \eta \cdot g^{(t)}$ using true gradient $g^{(t)} \in \mathbb{R}^d$, we apply sketch and de-sketch to the gradient:

$$w^{(t+1)} \leftarrow w^{(t)} - \eta \cdot R^\top \cdot R \cdot g^{(t)}.$$

Here $R \in \mathbb{R}^{b_{\text{sketch}} \times d}$ denotes a sketching matrix that sketches the true gradient to a lower dimension and $R^\top \in \mathbb{R}^{d \times b_{\text{sketch}}}$ denotes the de-sketching process that maps the sketched gradient back to the true gradient dimension. The coordinate-wise embedding property (Song & Yu, 2021) ensures $R^\top R g^{(t)}$ being an unbiased estimator of $g^{(t)}$ with bounded second moments, implying the new sketched gradient descent scheme preserving the original convergence properties. Hence, all clients will only communicate sketched gradients to the server, the server averages the sketched gradients and broacasts it back to all clients. Finally, each client de-sketches the received gradients and perform local updates. Since the sketching dimension is always small compared to original dimension, we save communication cost per iteration via Johnson-Lindenstrauss.

Though the communication problem has been addressed, such framework still faces privacy challenge: applying Johnson-Lindenstrauss "masks" the communicated gradients, but in order to give a provable privacy guarantee on this framework, we introduce additive low-dimensional Gaussian noises to make sure that the communicated vectors themselves are differential private.

We summarize the contributions in this work as follows:

**Framework contribution:** We propose a new federated learning framework that iteratively applies sketch and de-sketch matrices to gradients, which enjoys the following advantages:

- **Privacy**: it preserves the privacy via additive low-dimensional Gaussian noise.
- **Communication**: it reduces communication per round, since at each synchronization step, only a sketched gradient of lower-dimensional is communicated.
- **Convergence**: it preserves the convergence rate of vanilla local gradient descent method.

**Technical contribution:** Our analysis technique is also of independent interest in the relating areas:

- Unlike classical sketch-and-solve paradigm, our iterative sketch and de-sketch method can be combined with gradient-based methods and extended to broader optimization problems.
- We provide rigorous analysis on the impact of introducing sketching through coordinate-wise embedding, which can be generalized to other areas (Song & Yu, 2021).
- As a by-product, we give a novel linear convergence result of local GD under the strongly-convex and smooth scheme.

## 2 RELATED WORK

**Federated Learning**    Federated learning (FL) is an emerging framework in distributed deep learning. FL allows multiple parties or clients collaboratively train a model without data sharing. Fl let the local client perform most of the computation and a central sever update the model parameters through aggregation then transfers the parameters to local models (Dean et al., 2012; Shokri & Shmatikov, 2015; McMahan et al., 2016; 2017). In this way, the details of the data are not disclosed in between each party.

Unlike the standard parallel setting, FL has three unique challenge (Li et al., 2020a), including communication cost, data heterogeneity and client robustness. In our work, we focus on the first two challenges. The training data are massively distributed over an incredibly large number of devices, and the connection between the central server and a device is slow. A direct consequence is the slow communication, which motivated communication-efficient FL algorithm. Federated average (FedAvg) (McMahan et al., 2017) firstly addressed the communication efficiency problem by introducing a global model to aggregate local stochastic gradient descent updates. Later, different variations and adaptations have arisen. This encompasses a myriad of possible approaches, including developing better optimization algorithms (Wang et al., 2020a) and generalizing model to heterogeneous clients under special assumptions (Zhao et al., 2018; Kairouz et al., 2019; Li et al., 2021).

**Local GD and Local SGD** To seek the communication efficiency of Federated learning, local SGD has been proposed (Konečný et al., 2016; McMahan et al., 2017), where each client does a few SGD iterations locally before the server averages the local estimators. Different variants of local SGD algorithms has been explored, including with momentum (Yu et al., 2019b; Wang et al., 2018), with quantization (Basu et al., 2019; Reisizadeh et al., 2020), and with various variance-reduction methods (Liang et al., 2019; Karimireddy et al., 2020). Convergence analysis for local SGD mainly focuses on two regimes: identical data regime (Stich, 2018; Basu et al., 2019; Stich & Karimireddy, 2019; Haddadpour & Mahdavi, 2019; Khaled et al., 2020) and heterogeneous data regime (Jiang & Agrawal, 2018; Yu et al., 2019a; Basu et al., 2019; Haddadpour & Mahdavi, 2019; Khaled et al., 2019; 2020). In this work, we propose our framework based upon vanilla local GD and our analysis focus on the heterogeneous data regime.

**Sketching** Sketching has many applications in numerical linear, such as linear regression, low-rank approximation (Clarkson & Woodruff, 2013; Nelson & Nguyên, 2013; Meng & Mahoney, 2013; Boutsidis & Woodruff, 2014; Song et al., 2017; Andoni et al., 2018), distributed problems (Woodruff & Zhong, 2016; Boutsidis et al., 2016), reinforcement learning Wang et al. (2020b), tensor decomposition (Song et al., 2019), clustering (Esfandiari et al., 2021), cutting plane method (Jiang et al., 2020), generative adversarial networks (Xiao et al., 2018) and linear programming (Lee et al., 2019; Jiang et al., 2021; Song & Yu, 2021).

**Notations** For a positive integer $n$, we use $[n]$ to denote the set $\{1, 2, \cdots, n\}$. We use $\mathbb{E}[\cdot]$ to denote expectation (if it exists), and use $\Pr[\cdot]$ to denote probability. For a vector $x$, we use $\|x\|_2 := (\sum_{i=1}^n x_i^2)^{1/2}$ to denote its $\ell_2$ norm. We denote $1_{\{x=l\}}$ for $l \in \mathbb{R}$ to be the indicator function which equals to 1 if $x = l$ and 0 otherwise. Let $f : A \to B$ and $g : C \to A$ be two functions, we use $f \circ g$ to denote the composition of functions $f$ and $g$, i.e., for any $x \in C$, $(f \circ g)(x) = f(g(x))$. We denote $I_d$ to be the identity mapping.

## 3 PROBLEM SETUP

Consider a federated learning scenario with $N$ clients and corresponding local losses $f_c : \mathbb{R}^d \to \mathbb{R}$, our goal is to find

$$\min_{x \in \mathbb{R}^d} f(x) := \frac{1}{N} \sum_{c=1}^N f_c(x) \tag{1}$$

In this work, we consider the following classical convex and smooth setting for our objectives.

**Assumption 3.1.** *Assume that the set of minimizers of (1) is nonempty. Each $f_c$ is $\mu$-strongly convex for $\mu \geq 0$ and $L$-smooth. That is, for all $x, y \in \mathbb{R}^d$,*

$$\frac{\mu}{2}\|y - x\|_2^2 \leq f_c(y) - f_c(x) + \langle y - x, \nabla f_c(x) \rangle \leq \frac{L}{2}\|y - x\|_2^2.$$

*Note in the case $\mu = 0$, this assumption reduces back to convexity and smoothness.*

Apart from the above assumption, we allow local losses to have arbitrary heterogeneity. In other words, we allow $f_c$'s to be arbitrary functions.

## 4 OUR ALGORITHM

In this section, we propose a federated learning framework that addresses the communication efficiency issue. When the learning gradients are of high dimension, classical federated learning framework which sends the exact gradient could incur a heavy communication cost per round. Sketching technique, which emerges as an effective way to reduce the dimension of vector while preserving significant amount of information (Sarlós, 2006; Woodruff, 2014), is highly preferred in this setting. It enables us to compress the gradient vector into a lower dimension while preserving convergence rates, and greatly saves the communication cost per round.

Motivated by above discussion, we propose the iterative sketching-based federated learning algorithm, which builds upon vanilla local gradient descent: we start with a predetermined sequence of

independent sketching matrices shared across all clients. In each round, local clients accumulate and sketch its change over $K$ local steps, then transmit the low-dimensional sketch to the server. Server then averages the sketches and transmits them back to all clients. Upon receiving, each client de-sketches to update the local model.

---

**Algorithm 1** Iterative Sketching-based Federated Learning Algorithm with $K$ local steps

---

1: **procedure** ITERATIVESKETCHINGFL
2:     Each client initializes $w^0$ using the same set of random seed
3:     **for** $t = 1 \rightarrow T$ **do**                            $\triangleright$ $T$ denotes the total number of global steps
4:         /* Client */
5:         **parfor** $c = 1 \rightarrow N$ **do**                    $\triangleright$ $N$ denotes the total number of clients
6:             **if** $t = 1$ **then**
7:                 $u_c^{t,0} \leftarrow w^0$                                  $\triangleright$ $u_c^{t,0} \in \mathbb{R}^d$
8:             **else**
9:                 $u_c^{t,0} \leftarrow w^{t-1} + \mathsf{desk}_t(\Delta \widetilde{w}^{t-1})$     $\triangleright$ $\mathsf{desk}_t : \mathbb{R}^{b_{\text{sketch}}} \rightarrow \mathbb{R}^d$ de-sketch the change
10:             **end if**
11:             $w^t \leftarrow u_c^{t,0}$
12:             **for** $k = 1 \rightarrow K$ **do**
13:                 $u_c^{t,k} \leftarrow u_c^{t,k-1} - \eta_{\text{local}} \cdot \nabla f_c(w)|_{w=u_c^{t,k-1}}$
14:             **end for**
15:             $\Delta w_c(t) \leftarrow u_c^{t,K} - w^t$
16:             Client $c$ sends $\mathsf{sk}_t(\Delta w_c(t))$ to server       $\triangleright$ $\mathsf{sk}_t : \mathbb{R}^d \rightarrow \mathbb{R}^{b_{\text{sketch}}}$ sketch the change
17:         **end parfor**
18:         /* Server */
19:         $\Delta \widetilde{w}^t \leftarrow \eta_{\text{global}} \cdot \frac{1}{N} \sum_{c=1}^N \mathsf{sk}_t(\Delta w_c(t))$                     $\triangleright$ $\Delta \widetilde{w}^t \in \mathbb{R}^d$
20:         Server sends $\Delta \widetilde{w}^t$ to each client
21:     **end for**
22: **end procedure**

---

We highlight several distinct features of our algorithm:

• **Communication:** In each sync step, we only communicates a low-dimensional sketched gradients, indicating a smaller communication cost per round. This property is particularly valuable in a small-bandwidth setting.

• **De-sketch:**[i]: We emphasize that unlike the classical sketch-and-solve paradigm that decreases the problem dimension, our algorithm applies sketching in each round, combined with a de-sketching process which recovers back to the true gradient dimension.

• **Simple server task:**: Server only needs to do simple averaging, indicating no need of a trustworthy party as the server.

• **Decentralization:**: Our algorithm can be generalized to decentralized learning settings, where local clients can only communicate with neighboring nodes. In this case, it requires $O(\text{diam})$ rounds to propagate the sketched local changes, where diam is the diameter of the network graph.

### 4.1   $\mathsf{sk}/\mathsf{desk}$ VIA COORDINATE WISE EMBEDDING

In this section, we discuss the concrete realization of the $\mathsf{sk}_t/\mathsf{desk}_t$ operators in Algorithm 1 through random sketching matrices. Note we should require any processed gradient $\mathsf{desk}_t \circ \mathsf{sk}_t(g)$ to "be close" to the true gradient $g$ to avoid breaking the convergence property of the algorithm. To achieve this, we first introduce the following property for a broad family of sketching matrices, namely *coordinate-wise embedding* (Jiang et al., 2021; Song & Yu, 2021), that naturally connects with $\mathsf{sk}_t/\mathsf{desk}_t$ operators.

**Definition 4.1** ($a$-coordinate-wise embedding). *We say a randomized matrix $R \in \mathbb{R}^{b_{\text{sketch}} \times d}$ satisfying $a$-coordinate wise embedding if for any vector $g, h \in \mathbb{R}^d$, we have $\mathbb{E}_{R \sim \Pi}[h^\top R^\top R g] = h^\top g$ and $\mathbb{E}_{R \sim \Pi}[(h^\top R^\top R g)^2] \leq (h^\top g)^2 + \frac{a}{b_{\text{sketch}}} \|h\|_2^2 \cdot \|g\|_2^2$.*

---

[i]We elaborate the difference of our iterative sketch and de-sketch approach compared to classical sketch-and-solve approaches in Section 4.2.

In general, well-known sketching matrices have their coordinate-wise embedding parameter $a$ being a small constant (See Section D). Note that if we choose $h$ to be one-hot vector $e_i$, then the above conditions translate to $\mathbb{E}_{R \sim \Pi}[R^\top R g] = g$ and $\mathbb{E}_{R \sim \Pi}[\|R^\top R g\|_2^2] \leq (1 + a \cdot \frac{d}{b_{\text{sketch}}}) \cdot \|g\|_2^2$.

This implies that by choosing

$$\mathsf{sk}_t = R_t \in \mathbb{R}^{b_{\text{sketch}} \times d} \text{ (sketching)}, \qquad \mathsf{desk}_t = R_t^\top \in \mathbb{R}^{d \times b_{\text{sketch}}} \text{ (de-sketching)} \qquad (2)$$

for any iteration $t \geq 1$, where $R_t$'s are independent random matrices with sketching dimension $b_{\text{sketch}}$, we obtain an unbiased sketching/de-sketching scheme with bounded variance as state in the following Theorem 4.2.

**Theorem 4.2.** *Let* $\mathsf{sk}_t$ *and* $\mathsf{desk}_t$ *be defined by Eq. (2) using a sequence of independent sketching matrices* $R_t \in \mathbb{R}^{b_{\text{sketch}} \times d}$ *satisfying* $a$-*coordinate wise embedding property (Def. 4.1). Then the following properties hold:*

1. *Independence: For different iterations,* $(\mathsf{sk}_t, \mathsf{desk}_t)$*'s are independent of each other.*

2. *Linearity: Both* $\mathsf{sk}_t$ *and* $\mathsf{desk}_t$ *are linear operators.*

3. *Unbiased estimator: For any fixed vector* $h \in \mathbb{R}^d$*, it holds* $\mathbb{E}[\mathsf{desk}_t(\mathsf{sk}_t(h))] = h$.

4. *Bounded second moment: For any fixed vector* $h \in \mathbb{R}^d$*, it holds* $\mathbb{E}[\|\mathsf{desk}_t(\mathsf{sk}_t(h))\|_2^2] \leq (1 + a \cdot d/b_{\text{sketch}}) \cdot \|h\|_2^2$.

We will use the above property to instantiate the convergent proof and communication complexity in section 5. We remark that unlike traditional sketching matrix $R$, one can intuitively think of matrix $R^\top$ as a "de-sketch" matrix, it undoes sketching and recovers the sketched vector to original dimension.

## 4.2 SKETCH-AND-SOLVE VS SKETCH-AND-DE-SKETCH

In this section, we provide a brief discussion of the difference between classical sketch-and-solve paradigm (Clarkson & Woodruff, 2013; Woodruff, 2014) and our sketch-and-de-sketch scheme. The general idea of sketch-and-solve is to apply sketching matrix $R$ to the *entire problem*, and use certain black-box algorithm to solve the sketched-down low-dimensional version of the problem. Intuitively, sketching preserves the structure of the *problem*, therefore, the same algorithm can be exploited on a smaller problem, and to achieve a $1 \pm \epsilon$ guarantee. One downside of this method is the problem that is applicable needs to have certain structures. For example, it is not clear that given a non-convex objective, directly applying sketching will do any help.

In contrary, our sketch-and-de-sketch scheme only applies sketching to certain key component of a problem, for example, the gradient in a gradient descent method. Unlike sketch-and-solve, using sketch and de-sketch, we preserve the structure of the *algorithm*. This makes it feasible to a wider range of applications, such as showing the progress of gradient descent on non-convex objective, as we will show in appendix E and G. One drawback of this scheme is it gives a worse approximation guarantee $(1 \pm O(\frac{d}{b_{\text{sketch}}}))$, but in gradient descent, this can be mitigated via choosing a smaller stepsize for compensation. In a distributed setting, where privacy of communicated message and its size are main concerns, sketch-and-de-sketch is particularly valuable.

## 4.3 PRIVACY-PRESERVED SKETCHING VIA LOW-DIMENSIONAL NOISES

In this section, we discuss how to add low-dimensional Gaussian noise to make Algorithm 1 differential private. Consider a gradient vector $g_c$ generated via training of a client $c$, and another vector $g_c'$ that differs from $g_c$ by exactly one entry. Further, we assume this difference has a bounded magnitude of $\gamma$ in terms of absolute value. The goal is to protect the *sketched gradient*, $Rg_c$. Our strategy is as follows: on line 16 of Algorithm 1, we add a random vector $\Delta_c \in \mathbb{R}^{b_{\text{sketch}}}$ where each entry of $\Delta_c$ is drawn from $\mathcal{N}(0, \sigma^2)$ and with $\sigma \geq \Omega(\sqrt{d/b_{\text{sketch}}} \gamma \epsilon^{-1} \sqrt{\log(1/\delta)})$, then this modification on line 16 will produce an $(\epsilon, \delta)$-differential private guarantee for the sketched gradient.

We remark this is the most important step to preserve the privacy of the federated learning algorithm, since potential attackers might hack into a single client and have access to the $\mathsf{sk}_t/\mathsf{desk}_t$ operators for

each iteration. If they can further observe the communicated gradients from other clients, then the power of random masking with sketching is almost diminished, since they intuitively, they can de-sketch the sketched changes to obtain useful information. After adding low-dimensional Gaussian noises, attackers can no longer obtain useful information even they have the $\mathsf{sk}_t/\mathsf{desk}_t$ operators, which significantly improves the privacy of our system.

From an analytical perspective, adding this noise does not affect our analysis too much — since it's sampled from a zero-mean Gaussian distribution, our estimator $R^\top(Rg + \Delta_c)$ is still an unbiased estimator of vector $g$, it merely adds a variance term which can be factored into the variance of our un-modified estimator. Hence, in the analysis section, we present the theorems and proofs for the scenario where noises are not added. Similar analysis can be adapted to additive noise version.

## 5 CONVERGENCE THEORY AND COMMUNICATION COMPLEXITY

In this section, we analyze the convergence property of our proposed framework for smooth and convex objectives. Note the similarity shared by our framework and classical federated learning algorithms. We try to follow the existing analysis and focus on discussing the impact of the introduced randomness due to the sketching and de-sketching. We will show our approach enjoys benign convergence property and does not increase total communication at all.

### 5.1 SINGLE-STEP SCHEME

To start off, we consider a simple scenario where the number of local steps $K = 1$. In this case, by the linearity of $\mathsf{sk}_t$ and $\mathsf{desk}_t$ operators, the update rule can be written as

$$w^{t+1} = w^t - \eta \cdot \mathsf{desk}_t(\mathsf{sk}_t(\nabla f(w^t))). \tag{3}$$

We remark that in the case of $\mathsf{sk}_t$ and $\mathsf{desk}_t$ being identity mappings, our framework is equivalent to a distributed implementation of the vanilla gradient descent algorithm. Therefore, we follow the classical analysis of gradient descent and have the following key lemma:

**Lemma 5.1.** *If Assumption 3.1 holds and $K = 1$. Denote $\eta := \eta_{\text{local}} \cdot \eta_{\text{global}}$. Then we have*

$$\mathbb{E}[\|w^{t+1} - w^*\|_2^2] \leq (1 - \eta\mu)\,\mathbb{E}[\|w^t - w^*\|_2^2] - 2\eta(1 - \eta(1+\alpha)L) \cdot \mathbb{E}[f(w^t) - f(w^*)].$$

*where $w^*$ is a minimizer of problem (1).*

*Proof.* Note the update rule (3) implies

$$\|w^{t+1} - w^*\|_2^2 = \|w^t - w^*\|_2^2 - 2\eta\langle w^t - w^*, \mathsf{desk}_t(\mathsf{sk}_t(\nabla f(w^t)))\rangle + \eta^2\|\mathsf{desk}_t(\mathsf{sk}_t(\nabla f(w^t)))\|_2^2.$$

Taking the conditional expectation over last synchronization, we have by Theorem 4.2,

$$\mathbb{E}[\langle w^t - w^*, \mathsf{desk}_t(\mathsf{sk}_t(\nabla f(w^t)))\rangle \mid \mathcal{F}_t] = \langle w^t - w^*, \nabla f(w^t)\rangle,$$
$$\mathbb{E}[\|\mathsf{desk}_t(\mathsf{sk}_t(\nabla f(w^t)))\|_2^2 \mid \mathcal{F}_t] \leq (1 + \alpha)\|\nabla f(w^t)\|_2^2,$$

Combining with $\mu$-strongly convexity and $L$-smoothness of $f$, we obtain

$$\mathbb{E}[\|w^{t+1} - w^*\|_2^2 \mid \mathcal{F}_t] \leq \|w^t - w^*\|_2^2 - 2\eta\langle w^t - w^*, \nabla f(w^t)\rangle + \eta^2(1+\alpha)\|\nabla f(w^t)\|_2^2$$
$$\leq (1 - \eta\mu)\|w^t - w^*\|_2^2 - 2\eta(1 - \eta(1+\alpha)L) \cdot (f(w^t) - f(w^*)).$$

Taking the expectation of both sides over $\mathcal{F}_t$ we complete the proof. $\qquad\square$

Lemma 5.1 implies the introduced randomness from gradients only influence the second order term by a multiplicative factor $1+\alpha$ in expectation. Therefore, by scaling down the stepsize by a factor of $1+\alpha$, we can obtain the exact same convergence guarantee as the vanilla gradient descent algorithm. For the strongly convex and smooth objective case, we obtain the following linear convergence.

**Theorem 5.2.** *If Assumption 3.1 holds with $\mu > 0$. Let $K = 1$ and $\eta := \eta_{\text{local}} \cdot \eta_{\text{global}} \leq \frac{1}{(1+\alpha)L}$, then we have $\mathbb{E}[f(w^T) - f(w^*)] \leq \frac{L}{2}\,\mathbb{E}[\|w^0 - w^*\|_2^2]e^{-\eta\mu T}$, where $w^*$ is a minimizer of problem (1).*

*Proof.* By the choice of stepsize and Lemma 5.1, we obtain

$$\mathbb{E}[\|w^T - w^*\|_2^2] \leq (1 - \eta\mu)^T \mathbb{E}[\|w^0 - w^*\|_2^2].$$

Therefore, we have

$$\mathbb{E}[f(w^T) - f(w^*)] \leq \frac{L}{2} \mathbb{E}[\|w^T - w^*\|_2^2] \leq \frac{L}{2} \mathbb{E}[\|w^0 - w^*\|_2^2] e^{-\eta\mu T}. \qquad \square$$

For the convex case, we consider the average of iterations and obtain the sublinear convergence.

**Theorem 5.3.** *If Assumption 3.1 holds with $\mu = 0$. Let $K = 1$ and $\eta := \eta_{\text{local}} \cdot \eta_{\text{global}} \leq \frac{1}{2(1+\alpha)L}$, then we have $f(\overline{w}^T) - f(w^*) \leq \frac{1}{\eta(T+1)} \mathbb{E}[\|w^0 - w^*\|_2^2]$. where $w^*$ is a minimizer of problem (1) and $\overline{w}^T = \frac{1}{T+1} \sum_{t=0}^{T} w^t$.*

*Proof.* By the choice of stepsize and Lemma 5.1, we obtain

$$\eta(f(w^t) - f(w^*)) \leq \mathbb{E}[\|w^t - w^*\|_2^2] - \mathbb{E}[\|w^{t+1} - w^*\|_2^2].$$

Take the telescope summation over $t = 0, \cdots, T$, we have

$$\sum_{t=0}^{T}(f(w^t) - f(w^*)) \leq \frac{1}{\eta}(\mathbb{E}[\|w^0 - w^*\|_2^2] - \mathbb{E}[\|w^{T+1} - w^*\|_2^2]) \leq \frac{1}{\eta} \mathbb{E}[\|w^0 - w^*\|_2^2].$$

By the convexity of $f$ we complete the proof. $\qquad \square$

We point out that in the case of $\text{sk}_t$ and $\text{desk}_t$ being identity mappings, the parameter $\alpha$ reduces back 0 and Theorem 5.2 and 5.3 matches the convergence property of vanilla gradient descent exactly.

Further, above convergence results imply that comparing to vanilla gradient descent, our approach needs to shrink the stepsize by a factor of $O(\alpha)$, thus enlarge the number of iterations by a factor of $O(\alpha)$ to achieve desired accuracy. Therefore, using sketching matrices with dimension $b_{\text{sketch}}$, ours communicates $O(b_{\text{sketch}}/d \cdot \alpha)$ as many bits in total compared to vanilla approaches. According to Theorem 4.2, we have $\alpha = O(d/b_{\text{sketch}})$ for commonly used sketching matrices, implying our approach does not introduce extra communication cost at all.

## 5.2 MULTI-STEP SCHEME

Now we are ready to move on to general $K$ local step scheme. In this section, we assume $\eta_{\text{global}} = 1$. For notation simplicity, we denote $u_c^{t,-1} = u_c^{t-1,K-1}$ for $t \geq 2$. We also introduce the following notations of the average iterates, iterates variance, local gradients and average gradients to help with the analysis.

$$\overline{u}^{t,k} := \frac{1}{N} \sum_{c=1}^{N} u_c^{t,k}, \quad V^{t,k} := \frac{1}{N} \sum_{c=1}^{N} \|u_c^{t,k} - \overline{u}^{t,k}\|_2^2, \quad g_c^{t,k} := \nabla f_c(u_c^{t,k}), \quad \overline{g}^{t,k} := \frac{1}{N} \sum_{c=1}^{N} g_c^{t,k}$$

Using the new set of notations, one key observation is the average iterates satisfies

$$\overline{u}^{t,k} = \overline{u}^{t,k-1} - \eta_{\text{local}} \cdot \overline{g}^{t,k-1} + 1_{\{k=0\}} \cdot \eta_{\text{local}} \cdot (I_d - \text{desk}_t \circ \text{sk}_t)(\sum_{i=0}^{K-1} \overline{g}^{t-1,i}), \forall(t,k) \neq (1,0)$$

We remark again if $\text{sk}_t$ and $\text{desk}_t$ being identity mappings, our updates reduce back to the vanilla local gradient descent algorithm with $K$ local steps.

By considering the distance to optimal solution $\|\overline{u}^{t,k} - w^*\|_2^2$, we have the following intermediate lemma parallel to Lemma 5.1. Due to the space limitation, we defer the proof to appendix, see Lemma F.6.

**Lemma 5.4.** *If Assumption 3.1 holds and $\eta := \eta_{\text{local}} \leq \frac{1}{4L}$, then we have*

$$\mathbb{E}[\|\overline{u}^{t,k} - w^*\|_2^2] \leq (1 - \mu\eta) \mathbb{E}[\|\overline{u}^{t,k-1} - w^*\|_2^2] - \eta \mathbb{E}[f(\overline{u}^{t,k-1}) - f(w^*)] + 1.5\eta L \mathbb{E}[V^{t,k-1}]$$

$$+ 1_{\{k=0\}}\eta^2 \alpha K \cdot \left(4L \sum_{i=0}^{K-1} \mathbb{E}[f(\overline{u}^{t-1,i}) - f(w^*)] + 2L^2 \sum_{i=0}^{K-1} \mathbb{E}[V^{t-1,i}]\right)$$

*for any $(t,k) \neq (1,0)$, where $w^*$ is a minimizer of problem (1).*

Follow upon the above lemma, next step is to capture the quantity of the iterates variance $V^{t,k}$, we observe that in each round, we start off with $V^{t,0} = 0$ due to synchronization $u_c^{t,0} = \overline{u}^{t,0}$. Then $V^{t,k}$ can be viewed as the accumulation of the variance of the next $k$ local updates,

$$V^{t,k} = \frac{1}{N} \sum_{c=1}^{N} \|u_c^{t,0} - \sum_{i=0}^{k-1} \eta_{\text{local}} \cdot g_c^{t,i} - \overline{u}^{t,0} + \sum_{i=0}^{k-1} \eta_{\text{local}} \cdot \overline{g}^{t,i}\|_2^2 = \frac{\eta_{\text{local}}^2}{N} \sum_{c=1}^{N} \|\sum_{i=0}^{k-1} (\overline{g}^{t,i} - g_c^{t,i})\|_2^2$$

Therefore, it naturally requires us to characterize $V^{t,k}$ through certain measure of dissimilarity of local gradients. To achieve so, we avoid the common Lipschitz assumption which only gives a blur upper bound of the difference of local gradients, but follow the approach of Khaled et al. (2019; 2020), which focus on the quantity

$$\sigma^2 \overset{\text{def}}{=} \frac{1}{N} \sum_{c=1}^{N} \|\nabla f_c(w^*)\|^2 > 0$$

that is always finite and naturally characterize the degree of heterogeneity of local objectives. And we have the following observation of the summation of iterates variance over a single round:

**Lemma 5.5.** *If Assumption 3.1 holds and $\eta_{\text{local}} \leq \frac{1}{8LK}$. Then for any $t \geq 0$,*

$$\sum_{k=0}^{K-1} V^{t,k} \leq 8\eta_{\text{local}}^2 LK^2 \sum_{k=0}^{K-1} (f(\overline{u}^{t,k}) - f(w^*)) + 4\eta_{\text{local}}^2 K^3 \sigma^2$$

Combining Lemma 5.5 and Lemma 5.4, we notice that by choosing appropriately small stepsize, the first term in upper-bounding $V^{t,k}$ can be absorbed when we consider the average iterates over a single round. Therefore, We can obtain the following convergence result for the strongly convex and smooth losses.

**Theorem 5.6.** *If Assumption 3.1 holds with $\mu > 0$. If $\eta_{\text{local}} \leq \frac{1}{8(1+\alpha)LK}$,*

$$\mathbb{E}[f(w^{T+1}) - f(w^*)] \leq \frac{L}{2} \mathbb{E}[\|w^0 - w^*\|_2^2] e^{-\mu \eta_{\text{local}} T} + 4\eta_{\text{local}}^2 L^2 K^3 \sigma^2 / \mu.$$

*where $w^*$ is a minimizer of problem (1).*

*Proof.* Telescoping sum up Lemma 5.4 as $k$ varies from 0 to $K-1$ and absorb the higher-order terms by the choice of the stepsize, we have for any $t \geq 1$,

$$\mathbb{E}[\|\overline{u}^{t+1,0} - w^*\|_2^2] + \sum_{k=1}^{K-1} \mathbb{E}[\|\overline{u}^{t,k} - w^*\|_2^2] \leq (1 - \mu\eta_{\text{local}}) \sum_{k=0}^{K-1} \mathbb{E}[\|\overline{u}^{t,k} - w^*\|_2^2]) + 8\eta_{\text{local}}^3 LK^3 \sigma^2.$$

Rearranging the terms, we obtain for any $t \geq 1$,

$$\mathbb{E}[\|\overline{u}^{t+1,0} - w^*\|_2^2] \leq (1 - \mu\eta_{\text{local}}) \mathbb{E}[\|\overline{u}^{t,0} - w^*\|_2^2] + 8\eta_{\text{local}}^3 LK^3 \sigma^2,$$

implying

$$\mathbb{E}[\|w^{T+1} - w^*\|_2^2] \leq \mathbb{E}[\|w^0 - w^*\|_2^2] e^{-\mu\eta_{\text{local}} T} + 8\eta_{\text{local}}^2 LK^3 \sigma^2 / \mu.$$

We conclude by the $L$-smoothness of function $f$. $\qquad\square$

**Corollary 5.7.** *If Assumption 3.1 holds with $\mu > 0$. Then within Algorithm 1 outputs an $\epsilon$-optimal solution $w^T \in \mathbb{R}^d$ satisfying $\mathbb{E}[f(w^T) - f(w^*)] \leq \epsilon$ by using $O((LN/\mu) \max\{d, \sqrt{\sigma^2/(\mu\epsilon)}\} \log(L \mathbb{E}[\|w^0 - w^*\|_2^2]/\epsilon))$ bits of communication cost.*

We observe the same phenomenon as in the single-step scheme again, that compared to vanilla approaches, ours shrinks the stepsize by a factor of $O(\alpha)$, thus enlarge the number of rounds approximately by a factor of $O(\alpha)$. Since our approach only communicates $O(b_{\text{sketch}}/d)$ as many bits per round due to sketching, the total communication cost does not increase at all for commonly used sketching matrices, according to Theorem 4.2.

We also point out that when $\epsilon \geq \sigma^2/(\mu d^2)$, our analysis implies a linear convergence rate of local GD under only strongly-convex and smooth assumptions, which is new as far as we concern. We also have a similar observation in the convex losses case, as shown in the below theorem.

**Theorem 5.8.** *If Assumption 3.1 holds with $\mu = 0$. If $\eta_{\text{local}} \leq \frac{1}{8(1+\alpha)LK}$,*

$$\mathbb{E}[f(\overline{w}^T) - f(w^*)] \leq \frac{4\,\mathbb{E}[\|w^0 - w^*\|_2^2]}{\eta_{\text{local}}KT} + 32\eta_{\text{local}}^2 LK^2\sigma^2,$$

*where $\overline{w}^T = \frac{1}{KT}(\sum_{t=1}^{T}\sum_{k=0}^{K-1}\overline{u}^{t,k})$ is the average over parameters throughout the execution of Algorithm 1.*

*Proof.* Telescoping sum up Lemma 5.4 as $t$ varies from 0 to $T-1$ and $k$ varies from 0 to $K-1$ and absorb the higher-order terms by the choice of the stepsize,

$$\mathbb{E}[\|\overline{u}^{T+1,0} - w^*\|_2^2] - \mathbb{E}[\|w^0 - w^*\|_2^2] \leq -\frac{1}{4}\eta_{\text{local}}\sum_{t=1}^{T}\sum_{k=0}^{K-1}\mathbb{E}[f(\overline{u}^{t,k}) - f(w^*)] + 8\eta_{\text{local}}^3 LK^3 T\sigma^2.$$

Rearranging the above equation, we have

$$\frac{1}{KT}\sum_{t=1}^{T}\sum_{k=0}^{K-1}\mathbb{E}[f(\overline{u}^{t,k}) - f(w^*)] \leq \frac{4\,\mathbb{E}[\|w^0 - w^*\|_2^2]}{\eta_{\text{local}}KT} + 32\eta_{\text{local}}^2 LK^2\sigma^2.$$

By the convexity of $f$ we complete the proof. ◻

**Corollary 5.9.** *If Assumption 3.1 holds with $\mu = 0$. Then Algorithm 1 outputs an $\epsilon$-optimal solution $\overline{w}^T \in \mathbb{R}^d$ satisfying $\mathbb{E}[f(\overline{w}^T) - f(w^*)] \leq \epsilon$ by using $O(\mathbb{E}[\|w^0 - w^*\|_2^2]N\max\{Ld/\epsilon, \sigma\sqrt{L}/\epsilon^{3/2}\})$ bits of communication cost.*

We compare our communication cost with the work of Khaled et al. (2019), which analyzes the local gradient descent using the same assumption and framework. The result of Khaled et al. (2019) shows a communication cost of $O\left(\mathbb{E}[\|w^0 - w^*\|_2^2]Nd\max\{\frac{L}{\epsilon}, \frac{\sigma\sqrt{L}}{\epsilon^{3/2}}\}\right)$, which is strictly no less than our results. This shows again our approach does not introduce extra overall communication cost.

# 6    DISCUSSION

In this work, we propose the iterative sketch-based federated learning framework, which only communicates the sketched gradients with noises. Such framework enjoys the benefits of both better privacy and lower communication cost per round. We also rigorously prove that the randomness from sketching will not introduce extra overall communication cost.

Though our framework is built upon the local gradient descent algorithm and our theoretical discussion follows the analysis framework of Khaled et al. (2019; 2020), we emphasize that our approach and results can be extended to other gradient-based optimization algorithms and analysis, including but not limited to gradient descent with momentum and local stochastic gradient descent. The key reason is due to the sketched and de-sketched gradient $R^\top Rg$ is an unbiased estimator of the true gradient $g$ with second moments being a multiplier of $\|g\|_2^2$. As the iterates approach the optimal solution, the second moments approaches 0 correspondingly, resulting an exact match of the vanilla approach. Therefore, by scaling down the stepsize appropriately, we are able to recover the same convergence guarantee as the original gradient-based algorithms.

By a simple modification to our algorithm with additive Gaussian noises on the sketched gradients, we can also prove the differential privacy of our learning system by "hiding" the most important component in the system for guarding the safety and privacy. This additive noise also does not affect the convergence behavior of our algorithm too much, since it does make the estimator biased, and the additive variance can be factored into our original analysis.

Despite the benefits in terms of privacy and communication cost, we point out our approach does have trade-off on computation complexity. Since we need to shrink stepsize and thus enlarge the number of iterations to achieve certain accuracy level, our approach requires $O(d/b_{\text{sketch}})$ as many computational cost compared to vanilla approaches. However, in a privacy-and-communication focused distributed learning scenario, we hope this work provides a new solution and motivates future works.

**Ethics Statement.** This paper mainly focuses on the theoretical perspective of federated learning, and it proposes algorithm to improve the privacy of the learning system.

**Reproducibility Statement.** This paper contains several theoretical results, for discussions related to sketching, we refer readers to section D, for discussions related to the convergence analysis and communication costs of Algorithm 1, we refer readers to section F and G. For discussions related to differential privacy, we refer readers to section H.

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

## CONTENTS

**Roadmap.** We organize the appendix as follows. In section A, we introduce some notations and definitions that will be used across the appendix. In section B, we study several probability tools we will be using in the proof of cretain properties of various sketching matrices. In section C, we lay out some key assumptions on local objective function $f_c$ and global objective function $f$, in order to proceed our discussion of convergence theory. In section E, we give complete proofs for single-step scheme. We dedicate sections F and G to illustrate formal analysis of the convergence results of Algorithm 1 under $k$ local steps, given different assumptions of objective function $f$. In section H, we introduce additive noise to make our sketched gradients differential private and prove it for specific AMS sketch matrix.

## A  PRELIMINARY

For a positive integer $n$, we use $[n]$ to denote the set $\{1, 2, \cdots, n\}$. We use $\mathbb{E}[\cdot]$ to denote expectation (if it exists), and use $\Pr[\cdot]$ to denote probability. For a function $f$, we use $\widetilde{O}(f)$ to denote $f\operatorname{poly}(\log f)$. For a vector $x$, For a vector $x$, we use $\|x\|_1 := \sum_i |x_i|$ to denote its $\ell_1$ norm, we use $\|x\|_2 := (\sum_{i=1}^n x_i^2)^{1/2}$ to denote its $\ell_2$ norm, we use $\|x\|_\infty := \max_{i \in [n]} |x_i|$ to denote its $\ell_\infty$ norm. For a matrix $A$ and a vector $x$, we define $\|x\|_A := \sqrt{x^\top A x}$. For a full rank square matrix $A$, we use $A^{-1}$ to denote its true inverse. For a matrix $A$, we use $A^\dagger$ to denote its pseudo-inverse. For a matrix $A$, we use $\|A\|$ to denote its spectral norm. We use $\|A\|_F := (\sum_{i,j} A_{i,j}^2)^{1/2}$ to denote its Frobenius norm. We use $A^\top$ to denote the transpose of $A$. We denote $1_{\{x=l\}}$ for $l \in \mathbb{R}$ to be the indicator function which equals to 1 if $x = l$ and 0 otherwise. Let $f : A \to B$ and $g : C \to A$ be two functions, we use $f \circ g$ to denote the composition of functions $f$ and $g$, i.e., for any $x \in C$, $(f \circ g)(x) = f(g(x))$.

## B  PROBABILITY

**Lemma B.1** (Chernoff bound Chernoff (1952)). *Let $X = \sum_{i=1}^n X_i$, where $X_i = 1$ with probability $p_i$ and $X_i = 0$ with probability $1 - p_i$, and all $X_i$ are independent. Let $\mu = \mathbb{E}[X] = \sum_{i=1}^n p_i$. Then*
*1. $\Pr[X \geq (1+\delta)\mu] \leq \exp(-\delta^2 \mu/3), \forall \delta > 0$ ;*
*2. $\Pr[X \leq (1-\delta)\mu] \leq \exp(-\delta^2 \mu/2), \forall 0 < \delta < 1$.*

**Lemma B.2** (Hoeffding bound Hoeffding (1963)). *Let $X_1, \cdots, X_n$ denote $n$ independent bounded variables in $[a_i, b_i]$. Let $X = \sum_{i=1}^n X_i$, then we have*

$$\Pr[|X - \mathbb{E}[X]| \geq t] \leq 2\exp\left(-\frac{2t^2}{\sum_{i=1}^n (b_i - a_i)^2}\right).$$

**Lemma B.3** (Bernstein inequality Bernstein (1924)). *Let $X_1, \cdots, X_n$ be independent zero-mean random variables. Suppose that $|X_i| \leq M$ almost surely, for all $i$. Then, for all positive $t$,*

$$\Pr\left[\sum_{i=1}^n X_i > t\right] \leq \exp\left(-\frac{t^2/2}{\sum_{j=1}^n \mathbb{E}[X_j^2] + Mt/3}\right).$$

**Lemma B.4** (Khintchine's inequality, Khintchine (1923); Haagerup (1981)). *Let $\sigma_1, \cdots, \sigma_n$ be i.i.d. sign random variables, and let $z_1, \cdots, z_n$ be real numbers. Then there are constants $C > 0$ so that for all $t > 0$*

$$\Pr\left[\left|\sum_{i=1}^n z_i \sigma_i\right| \geq t\|z\|_2\right] \leq \exp(-Ct^2).$$

**Lemma B.5** (Hason-wright inequality Hanson & Wright (1971); Rudelson & Vershynin (2013)). *Let $x \in \mathbb{R}^n$ denote a random vector with independent entries $x_i$ with $\mathbb{E}[x_i] = 0$ and $|x_i| \leq K$. Let $A$ be an $n \times n$ matrix. Then, for every $t \geq 0$,*

$$\Pr[|x^\top A x - \mathbb{E}[x^\top A x]| > t] \leq 2 \cdot \exp(-c\min\{t^2/(K^4\|A\|_F^2), t/(K^2\|A\|)\}).$$

**Lemma B.6** (Lemma 1 on page 1325 of Laurent and Massart Laurent & Massart (2000)). *Let $X \sim \mathcal{X}_k^2$ be a chi-squared distributed random variable with $k$ degrees of freedom. Each one has zero mean and $\sigma^2$ variance. Then*

$$\Pr[X - k\sigma^2 \geq (2\sqrt{kt} + 2t)\sigma^2] \leq \exp(-t),$$

$$\Pr[k\sigma^2 - X \geq 2\sqrt{kt}\sigma^2] \leq \exp(-t).$$

**Lemma B.7** (Tail bound for sub-exponential distribution Foss et al. (2011)). *We say $X \in \mathrm{SE}(\sigma^2, \alpha)$ with parameters $\sigma > 0, \alpha > 0$ if:*

$$\mathbb{E}[e^{\lambda X}] \leq \exp(\lambda^2 \sigma^2 / 2), \quad \forall |\lambda| < 1/\alpha.$$

*Let $X \in \mathrm{SE}(\sigma^2, \alpha)$ and $\mathbb{E}[X] = \mu$, then:*

$$\Pr[|X - \mu| \geq t] \leq \exp(-0.5 \min\{t^2/\sigma^2, t/\alpha\}).$$

**Lemma B.8** (Matrix Chernoff bound Tropp (2011); Lu et al. (2013)). *Let $\mathcal{X}$ be a finite set of positive-semidefinite matrices with dimension $d \times d$, and suppose that*

$$\max_{X \in \mathcal{X}} \lambda_{\max}(X) \leq B.$$

*Sample $\{X_1, \cdots, X_n\}$ uniformly at random from $\mathcal{X}$ without replacement. We define $\mu_{\min}$ and $\mu_{\max}$ as follows:*

$$\mu_{\min} := n \cdot \lambda_{\min}(\underset{X \sim \mathcal{X}}{\mathbb{E}}[X]) \ \text{ and } \ \mu_{\max} := n \cdot \lambda_{\max}(\underset{X \sim \mathcal{X}}{\mathbb{E}}[X]).$$

*Then*

$$\Pr\left[\lambda_{\min}(\sum_{i=1}^{n} X_i) \leq (1-\delta)\mu_{\min}\right] \leq d \cdot \exp(-\delta^2 \mu_{\min}/B) \text{ for } \delta \in [0, 1),$$

$$\Pr\left[\lambda_{\max}(\sum_{i=1}^{n} X_i) \geq (1+\delta)\mu_{\max}\right] \leq d \cdot \exp\left(-\delta^2 \mu_{\max}/(4B)\right) \text{ for } \delta \geq 0.$$

## C  OPTIMIZATION BACKGROUNDS

**Definition C.1.** *Let $f : \mathbb{R}^d \to \mathbb{R}$ be a function, we say $f$ is $L$-smooth if for any $x, y \in \mathbb{R}^d$, we have*

$$\|\nabla f(x) - \nabla f(y)\|_2 \leq L\|x - y\|_2$$

*Equivalently, for any $x, y \in \mathbb{R}^d$, we have*

$$f(y) \leq f(x) + \langle y - x, \nabla f(x) \rangle + \frac{L}{2}\|y - x\|_2^2$$

**Definition C.2.** *Let $f : \mathbb{R}^d \to \mathbb{R}$ be a function, we say $f$ is convex if for any $x, y \in \mathbb{R}^d$, we have*

$$f(x) \geq f(y) + \langle x - y, \nabla f(y) \rangle$$

**Definition C.3.** *Let $f : \mathbb{R}^d \to \mathbb{R}$ be a function, we say $f$ is $\mu$-strongly-convex if for any $x, y \in \mathbb{R}^d$, we have*

$$\|\nabla f(x) - \nabla f(y)\|_2 \geq \mu\|x - y\|_2$$

*Equivalently, for any $x, y \in \mathbb{R}^d$, we have*

$$f(y) \geq f(x) + \langle y - x, \nabla f(x) \rangle + \frac{\mu}{2}\|y - x\|_2^2$$

**Fact C.4.** *Let $f : \mathbb{R}^d \to \mathbb{R}$ be an $L$-smooth and convex function, then for any $x, y \in \mathbb{R}^d$, we have*

$$f(y) - f(x) \geq \langle y - x, \nabla f(x) \rangle + \frac{1}{2L} \cdot \|\nabla f(y) - \nabla f(x)\|_2^2$$

**Fact C.5** (Inequality 4.12 in Bottou et al. (2018)). *Let $f : \mathbb{R}^d \to \mathbb{R}$ be a $\mu$-strongly convex function. Let $x^*$ be the minimizer of $f$. Then for any $x \in \mathbb{R}^d$, we have*

$$f(x) - f(x^*) \leq \frac{1}{2\mu}\|\nabla f(x)\|_2^2$$

# D  SKETCHING

In this section, we discuss the $(\alpha, \beta, \delta)$-coordinate wise embedding property we proposed in this work through several commonly used sketching matrices.

We consider several standard sketching matrices:

1. Random Gaussian matrices.

2. Subsampled randomized Hadamard/Fourier transform matrices Lu et al. (2013).

3. AMS sketch matrices Alon et al. (1999), random $\{-1, +1\}$ per entry.

4. Count-Sketch matrices Charikar et al. (2002), each column only has one non-zero entry, and is $-1, +1$ half probability each.

5. Sparse embedding matrices Nelson & Nguyên (2013), each column only has $s$ non-zero entries, and each entry is $-\frac{1}{\sqrt{s}}, +\frac{1}{\sqrt{s}}$ half probability each.

6. Uniform sampling matrices.

We list the definitions and results of above sketching matrices for coordinate-wise embedding in Table 1.

Table 1: Roadmap of the results for coordinate-wise embedding

| Sketching matrix | Definition | Expectation | Variance | Inner Product | Concentration |
|---|---|---|---|---|---|
| Random Gaussian | Definition D.2 | Lemma D.11 | Lemma D.13 | Lemma D.18 | Lemma D.24 |
| SRHT | Definition D.3 | Lemma D.11 | Lemma D.12 | Lemma D.19 | Lemma D.23 |
| AMS | Definition D.4 | Lemma D.11 | Lemma D.12 | Lemma D.20 | Lemma D.23 |
| Count-sketch | Definition D.5 | Lemma D.11 | Lemma D.14 | Lemma D.21 | Lemma D.25 |
| Sparse embedding | Definition D.6,D.7 | Lemma D.11 | Lemma D.15 | Lemma D.22 | Lemma D.28 |
| Uniform sampling | Definition D.8 | Lemma D.11 | Lemma D.16 | | Lemma D.29 |

## D.1  DEFINITION

**Definition D.1** ($k$-wise independence). $\mathcal{H} = \{h : [m] \to [l]\}$ *is a $k$-wise independent hash family if $\forall i_1 \neq i_2 \neq \cdots \neq i_k \in [n]$ and $\forall j_1, \cdots, j_k \in [l]$,*

$$\Pr_{h \in \mathcal{H}}[h(i_1) = j_1 \wedge \cdots \wedge h(i_k) = j_k] = \frac{1}{l^k}.$$

**Definition D.2** (Random Gaussian matrix). *We say $R \in \mathbb{R}^{b \times n}$ is a random Gaussian matrix if all entries are sampled from $\mathcal{N}(0, 1/b)$ independently.*

**Definition D.3** (Subsampled randomized Hadamard/Fourier transform matrix Lu et al. (2013)). *We say $R \in \mathbb{R}^{b \times n}$ is a subsampled randomized Hadamard transform matrix[ii] if it is of the form $R = \sqrt{n/b}SHD$, where $S \in \mathbb{R}^{b \times n}$ is a random matrix whose rows are $b$ uniform samples (without replacement) from the standard basis of $\mathbb{R}^n$, $H \in \mathbb{R}^{n \times n}$ is a normalized Walsh-Hadamard matrix, and $D \in \mathbb{R}^{n \times n}$ is a diagonal matrix whose diagonal elements are i.i.d. Rademacher random variables.*

**Definition D.4** (AMS sketch matrix Alon et al. (1999)). *Let $h_1, h_2, \cdots, h_b$ be $b$ random hash functions picking from a 4-wise independent hash family $\mathcal{H} = \{h : [n] \to \{-\frac{1}{\sqrt{b}}, +\frac{1}{\sqrt{b}}\}\}$. Then $R \in \mathbb{R}^{b \times n}$ is a AMS sketch matrix if we set $R_{i,j} = h_i(j)$.*

**Definition D.5** (Count-sketch matrix Charikar et al. (2002)). *Let $h : [n] \to [b]$ be a random 2-wise independent hash function and $\sigma : [n] \to \{-1, +1\}$ be a random 4-wise independent hash function. Then $R \in \mathbb{R}^{b \times n}$ is a count-sketch matrix if we set $R_{h(i),i} = \sigma(i)$ for all $i \in [n]$ and other entries to zero.*

---

[ii]In this case, we require $\log n$ to be an integer.

**Definition D.6** (Sparse embedding matrix I Nelson & Nguyên (2013))**.** *We say $R \in \mathbb{R}^{b \times n}$ is a sparse embedding matrix with parameter $s$ if each column has exactly $s$ non-zero elements being $\pm 1/\sqrt{s}$ uniformly at random, whose locations are picked uniformly at random without replacement (and independent across columns)* [iii].

**Definition D.7** (Sparse embedding matrix II Nelson & Nguyên (2013))**.** *Let $h : [n] \times [s] \to [b/s]$ be a a ramdom 2-wise independent hash function and $\sigma : [n] \times [s] \to \{-1, 1\}$ be a 4-wise independent. Then $R \in \mathbb{R}^{b \times n}$ is a sparse embedding matrix II with parameter $s$ if we set $R_{(j-1)b/s+h(i,j),i} = \sigma(i,j)/\sqrt{s}$ for all $(i,j) \in [n] \times [s]$ and all other entries to zero.* [iv]

**Definition D.8** (Uniform sampling matrix)**.** *We say $R \in \mathbb{R}^{b \times n}$ is a uniform sampling matrix if it is of the form $R = \sqrt{n/b}SD$, where $S \in \mathbb{R}^{b \times n}$ is a random matrix whose rows are $b$ uniform samples (without replacement) from the standard basis of $\mathbb{R}^n$, and $D \in \mathbb{R}^{n \times n}$ is a diagonal matrix whose diagonal elements are i.i.d. Rademacher random variables.*

### D.2 COORDINATE WISE EMBEDDING

We define coordinate-wise embedding as follows

**Definition D.9** (($\alpha, \beta, \delta$)-coordinate wise embedding)**.** *We say a randomized matrix $R \in \mathbb{R}^{b \times n}$ satisfying $(\alpha, \beta, \delta)$-coordinate wise embedding if*

1. $\underset{R \sim \Pi}{\mathbb{E}}[g^\top R^\top R h] = g^\top h$,

2. $\underset{R \sim \Pi}{\mathbb{E}}[(g^\top R^\top R h)^2] \leq (g^\top h)^2 + \frac{\alpha}{b}\|g\|_2^2 \|h\|_2^2$,

3. $\underset{R \sim \Pi}{\Pr}\left[|g^\top R^\top R h - g^\top h| \geq \frac{\beta}{\sqrt{b}}\|g\|_2 \|h\|_2\right] \leq \delta$.

**Remark D.10.** *Given a randomized matrix $R \in \mathbb{R}^{b \times n}$ satisfying $(\alpha, \beta, \delta)$-coordinate wise embedding and any orthogonal projection $P \in \mathbb{R}^{n \times n}$, above definition implies*

1. $\underset{R \sim \Pi}{\mathbb{E}}[PR^\top R h] = Ph$,

2. $\underset{R \sim \Pi}{\mathbb{E}}[(PR^\top R h)_i^2] \leq (Ph)_i^2 + \frac{\alpha}{b}\|h\|_2^2$,

3. $\underset{R \sim \Pi}{\Pr}\left[|(PR^\top R h)_i - (Ph)_i| \geq \frac{\beta}{\sqrt{b}}\|h\|_2\right] \leq \delta$.

*since $\|P\|_2 \leq 1$ implies $\|P_{i,:}\|_2 \leq 1$ for all $i \in [n]$.*

### D.3 EXPECTATION AND VARIANCE

**Lemma D.11.** *Let $R \in \mathbb{R}^{b \times n}$ denote any of the random matrix in Definition D.2, D.3, D.4, D.6, D.7, D.8. Then for any fixed vector $h \in \mathbb{R}^n$ and any fixed vector $g \in \mathbb{R}^n$, the following properties hold:*

$$\underset{R \sim \Pi}{\mathbb{E}}[g^\top R^\top R h] = g^\top h$$

*Proof.*

$$\underset{R \sim \Pi}{\mathbb{E}}[g^\top R^\top R h] = g^\top \underset{R \sim \Pi}{\mathbb{E}}[R^\top R]h = g^\top I h = g^\top h.$$

$\square$

**Lemma D.12.** *Let $R \in \mathbb{R}^{b \times n}$ denote a subsampled randomized Hadamard transform or AMS sketch matrix as in Definition D.3, D.4. Then for any fixed vector $h \in \mathbb{R}^n$ and any fixed vector $g \in \mathbb{R}^n$, the following properties hold:*

$$\underset{R \sim \Pi}{\mathbb{E}}[(g^\top R^\top R h)^2] \leq (g^\top h)^2 + \frac{2}{b}\|g\|_2^2 \cdot \|h\|_2^2.$$

---

[iii]For our purposes the signs need only be $O(\log d)$-wise independent, and each column can be specified by a $O(\log d)$-wise independent permutation, and the seeds specifying the permutations in different columns need only be $O(\log d)$-wise independent.

[iv]This definition has the same behavior as sparse embedding matrix I for our purpose.

*Proof.* If $\mathbb{E}_a[a] = b$, it is easy to see that

$$\mathbb{E}_a[(a-b)^2] = \mathbb{E}_a[a^2 - 2ab + b^2] = \mathbb{E}_a[a^2 - b^2]$$

We can rewrite it as follows:

$$\mathbb{E}_{R\sim\Pi}[(g^\top R^\top Rh)^2 - (g^\top h)^2] = \mathbb{E}_{R\sim\Pi}[(g^\top(R^\top R - I)h)^2],$$

It can be bounded as follows:

$$\mathbb{E}_{R\sim\Pi}[(g^\top(R^\top R - I)h)^2]$$

$$= \mathbb{E}_{R\sim\Pi}\left[\left(\sum_{k=1}^b (Rg)_k (Rh)_k - g^\top h\right)^2\right]$$

$$= \mathbb{E}_{R\sim\Pi}\left[\left(\sum_{k=1}^b \sum_{i=1}^n R_{k,i} g_i \cdot \sum_{j\in[n]\setminus\{i\}} R_{k,j} h_j\right)^2\right]$$

$$= \mathbb{E}_{R\sim\Pi}\left[\left(\sum_{k=1}^b \sum_{i=1}^n R_{k,i} g_i \cdot \sum_{j\in[n]\setminus\{i\}} R_{k,j} h_j\right) \cdot \left(\sum_{k'=1}^b \sum_{i'=1}^n R_{k',i'} g_{i'} \cdot \sum_{j'\in[n]\setminus\{i'\}} R_{k',j'} h_{j'}\right)\right]$$

$$= \mathbb{E}_{R\sim\Pi}\left[\left(\sum_{k=1}^b \sum_{i=1}^n R_{k,i}^2 g_i^2 \cdot \sum_{j\in[n]\setminus\{i\}} R_{k,j}^2 h_j^2\right) + \left(\sum_{k=1}^b \sum_{i=1}^n R_{k,i}^2 g_i h_i \cdot \sum_{j\in[n]\setminus\{i\}} R_{k,j}^2 g_j h_j\right)\right]$$

$$= \frac{1}{b}\left(\sum_{i=1}^n g_i^2 \sum_{j\in[n]\setminus\{i\}} h_j^2\right) + \frac{1}{b}\left(\sum_{i=1}^n g_i h_i \sum_{j\in[n]\setminus\{i\}} g_j h_j\right)$$

$$\leq \frac{2}{b}\|g\|_2^2\|h\|_2^2,$$

where the second step follows from $R_{k,i}^2 = 1/b$, $\forall k, i \in [b] \times [n]$, the forth step follows from $\mathbb{E}[R_{k,i}R_{k,j}R_{k',i'}R_{k',j'}] \neq 0$ only if $i = i'$, $j = j'$, $k = k'$ or $i = j'$, $j = i'$, $k = k'$, the fifth step follows from $R_{k,i}$ and $R_{k,j}$ are independent if $i \neq j$ and $R_{k,i}^2 = R_{k,j}^2 = 1/b$, and the last step follows from Cauchy-Schwartz inequality.

Therefore,

$$\mathbb{E}_{R\sim\Pi}[(g^\top R^\top Rh)^2 - (g^\top h)^2] = \mathbb{E}_{R\sim\Pi}[(g^\top(R^\top R - I)h)^2] \leq \frac{2}{b}\|g\|_2^2\|h\|_2^2.$$

$\square$

**Lemma D.13.** *Let $R \in \mathbb{R}^{b\times n}$ denote a random Gaussian matrix as in Definition D.2. Then for any fixed vector $h \in \mathbb{R}^n$ and any fixed vector $g \in \mathbb{R}^n$, the following properties hold:*

$$\mathbb{E}_{R\sim\Pi}[(g^\top R^\top Rh)^2] \leq (g^\top h)^2 + \frac{3}{b}\|g\|_2^2 \cdot \|h\|_2^2.$$

*Proof.* Note

$$\mathbb{E}_{R\sim\Pi}[(g^\top R^\top Rh)^2]$$

$$= \mathbb{E}_{R\sim\Pi}\left[\left(\sum_{k=1}^b \sum_{i=1}^n R_{k,i} g_i \cdot \sum_{j=1}^n R_{k,j} h_j\right)^2\right]$$

$$= \mathbb{E}_{R\sim\Pi}\left[\left(\sum_{k=1}^b \sum_{i=1}^n R_{k,i} g_i \cdot \sum_{j=1}^n R_{k,j} h_j\right) \cdot \left(\sum_{k'=1}^b \sum_{i'=1}^n R_{k',i'} g_{i'} \cdot \sum_{j'=1}^n R_{k',j'} h_{j'}\right)\right]$$

$$
\begin{aligned}
= \underset{R \sim \Pi}{\mathbb{E}} \Bigg[ & \left( \sum_{k=1}^{b} \sum_{k' \in [b] \setminus \{k\}} \sum_{i=1}^{n} \sum_{i'=1}^{n} R_{k,i}^2 R_{k',i'}^2 g_i h_i g_{i'} h_{i'} \right) + \left( \sum_{k=1}^{b} \sum_{i=1}^{n} R_{k,i}^4 g_i^2 h_i^2 \right) \\
& + \left( \sum_{k=1}^{b} \sum_{i=1}^{n} \sum_{j \in [n] \setminus \{i\}} R_{k,i}^2 R_{k,j}^2 g_i^2 h_j^2 \right) + \left( \sum_{k=1}^{n} \sum_{i=1}^{n} \sum_{i' \in [n] \setminus \{i\}} R_{k,i}^2 R_{k,i'}^2 g_i h_i g_{i'} h_{i'} \right) \\
& + \left( \sum_{k=1}^{b} \sum_{i=1}^{n} \sum_{j \in [n] \setminus \{i\}} R_{k,i}^2 R_{k,j}^2 g_i h_j g_j h_i \right) \Bigg] \\
= \frac{b-1}{b} & \sum_{i=1}^{n} \sum_{i'=1}^{n} g_i h_i g_{i'} h_{i'} + \frac{3}{b} \sum_{i=1}^{n} g_i^2 h_i^2 \\
& + \frac{1}{b} \sum_{i=1}^{n} \sum_{j \in [n] \setminus [i]} g_i^2 h_j^2 + \frac{1}{b} \sum_{i=1}^{n} \sum_{i' \in [n] \setminus [i]} g_i h_i g_{i'} h_{i'} + \frac{1}{b} \sum_{i=1}^{n} \sum_{j \in [n] \setminus [i]} g_i h_j g_j h_i \\
\leq (g^\top h)^2 & + \frac{3}{b} \|g\|_2^2 \|h\|_2^2,
\end{aligned}
$$

where the third step follows from that for independent entries of a random Gaussian matrix, $\mathbb{E}[R_{k,i} R_{k,j} R_{k',i'} R_{k',j'}] \neq 0$ only if 1. $k \neq k'$, $i = j$, $i' = j'$, or 2. $k = k'$, $i = i' = j = j'$, or 3. $k = k'$, $i = i' \neq j = j'$, or 4. $k = k'$, $i = j \neq i' = j'$, or 5. $k = k'$, $i = j' \neq i' = j$, the fourth step follows from $\mathbb{E}[R_{k,i}^2] = 1/b$ and $\mathbb{E}[R_{k,i}^4] = 3/b^2$, and the last step follows from Cauchy-Schwartz inequality. $\qquad \square$

**Lemma D.14.** *Let $R \in \mathbb{R}^{b \times n}$ denote a count-sketch matrix as in Definition D.5. Then for any fixed vector $h \in \mathbb{R}^n$ and any fixed vector $g \in \mathbb{R}^n$, the following properties hold:*

$$
\underset{R \sim \Pi}{\mathbb{E}}[(g^\top R^\top R h)^2] \leq (g^\top h)^2 + \frac{3}{b} \|g\|_2^2 \|h\|_2^2.
$$

*Proof.* Note

$$
\begin{aligned}
& \underset{R \sim \Pi}{\mathbb{E}}[(g^\top R^\top R h)^2] \\
= & \underset{R \sim \Pi}{\mathbb{E}} \left[ \left( \sum_{k=1}^{b} \sum_{i=1}^{n} R_{k,i} g_i \sum_{j=1}^{n} R_{k,j} h_j \right)^2 \right] \\
= & \underset{R \sim \Pi}{\mathbb{E}} \left[ \left( \sum_{k=1}^{b} \sum_{i=1}^{n} R_{k,i} g_i \sum_{j=1}^{n} R_{k,j} h_j \right) \cdot \left( \sum_{k'=1}^{b} \sum_{i'=1}^{n} R_{k',i'} g_{i'} \sum_{j'=1}^{n} R_{k',j'} h_{j'} \right) \right] \\
= & \underset{R \sim \Pi}{\mathbb{E}} \Bigg[ \left( \sum_{k=1}^{b} \sum_{k' \in [b] \setminus \{k\}} \sum_{i=1}^{n} \sum_{i' \in [n] \setminus \{i\}} R_{k,i}^2 R_{k',i'}^2 g_i h_i g_{i'} h_{i'} \right) + \left( \sum_{k=1}^{b} \sum_{i=1}^{n} R_{k,i}^4 g_i^2 h_i^2 \right) \\
& + \left( \sum_{k=1}^{b} \sum_{i=1}^{n} \sum_{j \in [n] \setminus \{i\}} R_{k,i}^2 R_{k,j}^2 g_i^2 h_j^2 \right) + \left( \sum_{k=1}^{n} \sum_{i=1}^{n} \sum_{i' \in [n] \setminus \{i\}} R_{k,i}^2 R_{k,i'}^2 g_i h_i g_{i'} h_{i'} \right) \\
& + \left( \sum_{k=1}^{b} \sum_{i=1}^{n} \sum_{j \in [n] \setminus \{i\}} R_{k,i}^2 R_{k,j}^2 g_i h_j g_j h_i \right) \Bigg] \\
= & \frac{b-1}{b} \sum_{i=1}^{n} \sum_{i' \in [n] \setminus i} g_i h_i g_{i'} h_{i'} + \sum_{i=1}^{n} g_i^2 h_i^2 \\
& + \frac{1}{b} \sum_{i=1}^{n} \sum_{j \in [n] \setminus \{i\}} g_i^2 h_j^2 + \frac{1}{b} \sum_{i=1}^{n} \sum_{i' \in [n] \setminus \{i\}} g_i h_i g_{i'} h_{i'} + \frac{1}{b} \sum_{i=1}^{n} \sum_{j \in [n] \setminus \{i\}} g_i h_j g_j h_i
\end{aligned}
$$

$$\leq (g^\top h)^2 + \frac{3}{b}\|g\|_2^2\|h\|_2^2,$$

where in the third step we are again considering what values of $k, k', i, i', j, j'$ that makes $\mathbb{E}[R_{k,i}R_{k,j}R_{k',i'}R_{k',j'}] \neq 0$. Since the hash function $\sigma(\cdot)$ of the count-sketch matrix is 4-wise independent, $\forall k, k'$, when $i \neq i' \neq j \neq j'$, or $i = i' = j \neq j'$ (and the other 3 symmetric cases), we have that $\mathbb{E}[R_{k,i}R_{k,j}R_{k',i'}R_{k',j'}] = 0$. Since the count-sketch matrix has only one non-zero entry in every column, when $k \neq k'$, if $i = i'$ or $i = j'$ or $j = i'$ or $j = j'$, we also have $\mathbb{E}[R_{k,i}R_{k,j}R_{k',i'}R_{k',j'}] = 0$. Thus we only need to consider the cases: 1. $k \neq k'$, $i = j \neq i' = j'$, or 2. $k = k'$, $i = i' = j = j'$, or 3. $k = k'$, $i = i' \neq j = j'$, or 4. $k = k'$, $i = j \neq i' = j'$, or 5. $k = k'$, $i = j' \neq i' = j$. And the fourth step follows from $\mathbb{E}[R_{k,i}^2] = 1/b$ and $\mathbb{E}[R_{k,i}^4] = 1/b$, and the last step follows from Cauchy-Schwartz inequality. $\qquad\square$

**Lemma D.15.** *Let $R \in \mathbb{R}^{b \times n}$ denote a sparse embedding matrix as in Definition D.6, D.7. Then for any fixed vector $h \in \mathbb{R}^n$ and any fixed vector $g \in \mathbb{R}^n$, the following properties hold:*

$$2. \underset{R \sim \Pi}{\mathbb{E}}[(g^\top R^\top Rh)^2] \leq (g^\top h)^2 + \frac{2}{b}\|g\|_2^2 \cdot \|h\|_2^2.$$

*Proof.* Note

$$\underset{R \sim \Pi}{\mathbb{E}}[(g^\top R^\top Rh)^2]$$

$$= \underset{R \sim \Pi}{\mathbb{E}}\left[\left(\sum_{k=1}^b \sum_{i=1}^n R_{k,i}g_i \sum_{j=1}^n R_{k,j}h_j\right)^2\right]$$

$$= \underset{R \sim \Pi}{\mathbb{E}}\left[\left(\sum_{k=1}^b \sum_{i=1}^n R_{k,i}g_i \sum_{j=1}^n R_{k,j}h_j\right) \cdot \left(\sum_{k'=1}^b \sum_{i'=1}^n R_{k',i'}g_{i'} \sum_{j'=1}^n R_{k',j'}h_{j'}\right)\right]$$

$$= \underset{R \sim \Pi}{\mathbb{E}}\Bigg[\left(\sum_{k=1}^b \sum_{i=1}^n R_{k,i}^2 g_i^2 \sum_{j \in [n]\setminus\{i\}} R_{k,j}^2 h_j^2\right) + \left(\sum_{k=1}^b \sum_{i=1}^n R_{k,i}^2 g_i h_i \sum_{j \in [n]\setminus\{i\}} R_{k,j}^2 g_j h_j\right)$$

$$+ \left(\sum_k \sum_{i \neq i'} R_{k,i}^2 R_{k,i'}^2 g_i h_i g_{i'} h_{i'}\right) + \left(\sum_k \sum_i R_{k,i}^4 g_i^2 h_i^2\right) + \left(\sum_{k \neq k'} \sum_{i \neq i'} R_{k,i}^2 R_{k',i'}^2 g_i h_i g_{i'} h_{i'}\right)$$

$$+ \left(\sum_{k \neq k'} \sum_i R_{k,i}^2 R_{k',i}^2 g_i^2 h_i^2\right)\Bigg]$$

$$= \frac{1}{b}\sum_{i \neq j} g_i^2 h_j^2 + \frac{1}{b}\sum_{i \neq j} g_i h_i g_j h_j + \frac{1}{b}\sum_{i \neq i'} g_i h_i g_{i'} h_{i'} + \frac{1}{s}\sum_i g_i^2 h_i^2 + \frac{b-1}{b}\sum_{i \neq i'} g_i h_i g_{i'} h_{i'} + \frac{s-1}{s}\sum_i g_i^2 h_i^2$$

$$\leq (g^\top h)^2 + \frac{2}{b}\|g\|_2^2\|h\|_2^2,$$

where the third step follows from the fact that the sparse embedding matrix has independent columns and $s$ non-zero entry in every column, the fourth step follows from $\mathbb{E}[R_{k,i}^2] = 1/b$, $\mathbb{E}[R_{k,i}^4] = 1/(bs)$, and $\mathbb{E}[R_{k,i}^2 R_{k',i}^2] = \frac{s(s-1)}{b(b-1)} \cdot \frac{1}{s^2}, \forall k \neq k'$ and the last step follows from Cauchy-Schwartz inequality. $\qquad\square$

**Lemma D.16.** *Let $R \in \mathbb{R}^{b \times n}$ denote a uniform sampling matrix as in Definition D.8. Then for any fixed vector $h \in \mathbb{R}^n$ and any fixed vector $g \in \mathbb{R}^n$, the following properties hold:*

$$2. \underset{R \sim \Pi}{\mathbb{E}}[(g^\top R^\top Rh)^2] \leq (g^\top h)^2 + \frac{n}{b}\|g\|_2^2\|h\|_2^2.$$

*Proof.* Note

$$\underset{R \sim \Pi}{\mathbb{E}}[(g^\top R^\top Rh)^2]$$

$$
\begin{aligned}
&= \mathop{\mathbb{E}}_{R \sim \Pi} \left[ \left( \sum_{k=1}^{b} \sum_{i=1}^{n} R_{k,i} g_i \sum_{j=1}^{n} R_{k,j} h_j \right)^2 \right] \\
&= \mathop{\mathbb{E}}_{R \sim \Pi} \left[ \left( \sum_{k=1}^{b} \sum_{i=1}^{n} R_{k,i} g_i \sum_{j=1}^{n} R_{k,j} h_j \right) \cdot \left( \sum_{k'=1}^{b} \sum_{i'=1}^{n} R_{k',i'} g_{i'} \sum_{j'=1}^{n} R_{k',j'} h_{j'} \right) \right] \\
&= \mathop{\mathbb{E}}_{R \sim \Pi} \left[ \left( \sum_{k} \sum_{i} R_{k,i}^4 g_i^2 h_i^2 \right) + \left( \sum_{k \neq k'} \sum_{i \neq i'} R_{k,i}^2 R_{k',i'}^2 g_i h_i g_{i'} h_{i'} \right) \right] \\
&= \frac{n}{b} \sum_{i} g_i^2 h_i^2 + \frac{(b-1)n}{(n-1)b} \sum_{i \neq i'} g_i h_i g_{i'} h_{i'} \\
&\leq (g^\top h)^2 + \frac{n}{b} \|g\|_2^2 \|h\|_2^2,
\end{aligned}
$$

where the third step follows from the fact that the random sampling matrix has one non-zero entry in every row, the fourth step follows from $\mathbb{E}[R_{k,i}^2 R_{k',i'}^2] = n/((n-1)b^2)$ for $k \neq k'$, $i \neq i'$ and $\mathbb{E}[R_{k,i}^4] = n/b^2$. □

**Remark D.17.** *Lemma D.16 indicates that uniform sampling fails in bounding variance in some sense, since the upper bound give here involves $n$.*

### D.4  BOUNDING INNER PRODUCT

**Lemma D.18** (Gaussian). *Let $R \in \mathbb{R}^{b \times n}$ be a random Gaussian matrix (Definition D.2). Then we have:*

$$
\Pr \left[ \max_{i \neq j} |\langle R_{*,i}, R_{*,j} \rangle| \geq \frac{\sqrt{\log(n/\delta)}}{\sqrt{b}} \right] \leq \Theta(\delta).
$$

*Proof.* Note for $i \neq j$, $R_{*,i}, R_{*,j} \sim \mathcal{N}(0, \frac{1}{b} I_b)$ are two independent Gaussian vectors. Let $z_k = R_{k,i} R_{k,j}$ and $z = \langle R_{*,i}, R_{*,j} \rangle$. Then we have for any $|\lambda| \leq b/2$,

$$
\mathbb{E}[e^{\lambda z_k}] = \frac{1}{\sqrt{1 - \lambda^2/b^2}} \leq \exp(\lambda^2/b^2),
$$

where the first step follows from $z_k = \frac{1}{4}(R_{k,i} + R_{k,j})^2 + \frac{1}{4}(R_{k,i} - R_{k,j})^2 = \frac{b}{2}(Q_1 - Q_2)$ where $Q_1, Q_2 \sim \chi_1^2$, and $\mathbb{E}[e^{\lambda Q}] = \frac{1}{\sqrt{1-2\lambda}}$ for any $Q \sim \chi_1^2$.

This implies $z_k \in \mathrm{SE}(2/b^2, 2/b)$ is a sub-exponential random variable. Thus, we have $z = \sum_{k=1}^{b} z_k \in \mathrm{SE}(2/b, 2/b)$, by sub-exponential concentration Lemma B.7 we have

$$
\Pr[|z| \geq t] \leq 2 \exp(-bt^2/4)
$$

for $0 < t < 1$. Picking $t = \sqrt{\log(n^2/\delta)/b}$, we have

$$
\Pr \left[ |\langle R_{*,i}, R_{*,j} \rangle| \geq \frac{c\sqrt{\log(n/\delta)}}{\sqrt{b}} \right] \leq \delta/n^2.
$$

Taking the union bound over all $(i,j) \in [n] \times [n]$ and $i \neq j$, we complete the proof. □

**Lemma D.19** (SRHT). *Let $R \in \mathbb{R}^{b \times n}$ be a subsample randomized Hadamard transform (Definition D.3). Then we have:*

$$
\Pr \left[ \max_{i \neq j} |\langle R_{*,i}, R_{*,j} \rangle| \geq \frac{\sqrt{\log(n/\delta)}}{\sqrt{b}} \right] \leq \Theta(\delta).
$$

*Proof.* For fixed $i \neq j$, let $X = [R_{*,i}, R_{*,j}] \in \mathbb{R}^{b \times 2}$. Then $X^\top X = \sum_{k=1}^{b} G_k$, where

$$G_k = [R_{k,i}, R_{k,j}]^\top [R_{k,i}, R_{k,j}] = \begin{bmatrix} \frac{1}{b} & R_{k,i} R_{k,j} \\ R_{k,i} R_{k,j} & \frac{1}{b} \end{bmatrix}.$$

Note the eigenvalues of $G_k$ are 0 and $\frac{2}{b}$ and $\mathbb{E}[X^\top X] = b \cdot \mathbb{E}[G_k] = I_2$ for all $k \in [b]$. Thus, applying matrix Chernoff bound B.8 to $X^\top X$ we have

$$\Pr\left[\lambda_{\max}(X^\top X) \leq 1 - t\right] \leq 2 \exp\left(-t^2 b / 2\right) \text{ for } t \in [0, 1), \text{ and}$$

$$\Pr\left[\lambda_{\max}(X^\top X) \geq 1 + t\right] \leq 2 \exp\left(-t^2 b / 8\right) \text{ for } t \geq 0.$$

which implies the eigenvalues of $X^\top X$ are between $[1-t, 1+t]$ with probability $1 - 4 \exp\left(-\frac{t^2 b}{8}\right)$. So the eigenvalues of $X^\top X - I_2$ are between $[-t, t]$ with probability $1 - 4 \exp\left(-\frac{t^2 b}{8}\right)$. Picking $t = \frac{c\sqrt{\log(n/\delta)}}{\sqrt{b}}$, we have

$$\Pr\left[\|X^\top X - I_2\| \geq \frac{c\sqrt{\log(n/\delta)}}{\sqrt{b}}\right] \leq \frac{\delta}{n^2}.$$

Note

$$X^\top X - I_2 = \begin{bmatrix} 0 & \langle R_{*,i}, R_{*,j} \rangle \\ \langle R_{*,i}, R_{*,j} \rangle & 0 \end{bmatrix},$$

whose spectral norm is $|\langle R_{*,i}, R_{*,j} \rangle|$. Thus, we have

$$\Pr\left[|\langle R_{*,i}, R_{*,j} \rangle| \geq \frac{c\sqrt{\log(n/\delta)}}{\sqrt{b}}\right] \leq \delta/n^2.$$

Taking a union bound over all pairs $(i, j) \in [n] \times [n]$ and $i \neq j$, we complete the proof. $\square$

**Lemma D.20** (AMS)**.** *Let $R \in \mathbb{R}^{b \times n}$ be a random AMS matrix (Definition D.4). Let $\{\sigma_i,\ i \in [n]\}$ be independent Rademacher random variables and $\overline{R} \in \mathbb{R}^{b \times n}$ with $\overline{R}_{*,i} = \sigma_i R_{*,i}, \ \forall i \in [n]$. Then we have:*

$$\Pr\left[\max_{i \neq j} |\langle \overline{R}_{*,i}, \overline{R}_{*,j} \rangle| \geq \frac{\sqrt{\log(n/\delta)}}{\sqrt{b}}\right] \leq \Theta(\delta).$$

*Proof.* Note for any fixed $i \neq j$, $\overline{R}_{*,i}$ and $\overline{R}_{*,j}$ are independent. By Hoeffding inequality (Lemma B.2), we have

$$\Pr\left[|\langle \overline{R}_{*,i}, \overline{R}_{*,j} \rangle| \geq t\right] \leq 2 \exp\left(-\frac{2t^2}{\sum_{i=1}^{b}(\frac{1}{b} - (-\frac{1}{b}))^2}\right) \leq 2 e^{-t^2 b / 2}$$

Choosing $t = \sqrt{2\log(2n^2/\delta)}/\sqrt{b}$, we have

$$\Pr\left[|\langle \overline{R}_{*,i}, \overline{R}_{*,j} \rangle| \geq \sqrt{2\log(2n^2/\delta)}/\sqrt{b}\right] \leq \frac{\delta}{n^2}.$$

Taking a union bound over all pairs $(i, j) \in [n] \times [n]$ and $i \neq j$, we complete the proof. $\square$

**Lemma D.21** (Count-Sketch)**.** *Let $R \in \mathbb{R}^{b \times n}$ be a count-sketch matrix (Definition D.5). Let $\{\sigma_i,\ i \in [n]\}$ be independent Rademacher random variables and $\overline{R} \in \mathbb{R}^{b \times n}$ with $\overline{R}_{*,i} = \sigma_i R_{*,i}, \ \forall i \in [n]$. Then we have:*

$$\max_{i \neq j} |\langle \overline{R}_{*,i}, \overline{R}_{*,j} \rangle| \leq 1.$$

*Proof.* Directly follow the definition of count-sketch matrices. $\square$

**Lemma D.22** (Sparse embedding). *Let $R \in \mathbb{R}^{b \times n}$ be a sparse embedding matrix with parameter $s$ (Definition D.6 and D.7). Let $\{\sigma_i, \ i \in [n]\}$ be independent Rademacher random variables and $\overline{R} \in \mathbb{R}^{b \times n}$ with $\overline{R}_{*,i} = \sigma_i R_{*,i}, \ \forall i \in [n]$. Then we have:*

$$\Pr\left[\max_{i \neq j}|\langle \overline{R}_{*,i}, \overline{R}_{*,j}\rangle| \geq \frac{c\sqrt{\log(n/\delta)}}{\sqrt{s}}\right] \leq \Theta(\delta).$$

*Proof.* Note for fixed $i \neq j$, $\overline{R}_{*,i}$ and $\overline{R}_{*,j}$ are independent. Assume $R_{*,i}$ and $R_{*,j}$ has $u$ non-zero elements at the same positions, where $0 \leq u \leq s$, then by Hoeffding inequality (Lemma B.2), we have

$$\Pr[|\langle \overline{R}_{*,i}, \overline{R}_{*,j}\rangle| \geq t] \leq 2\exp\left(-\frac{2t^2}{\sum_{i=1}^{u}(\frac{1}{s} - (-\frac{1}{s}))^2}\right) \leq 2\exp(-t^2 s^2/(2u)) \tag{4}$$

Let $t = \sqrt{(2u/s^2)\log(2n^2/\delta)}$, we have

$$\Pr\left[|\langle \overline{R}_{*,i}, \overline{R}_{*,j}\rangle| \geq \sqrt{2s^{-1}\log(2n^2/\delta)}\right] \leq \Pr\left[|\langle R_{*,i}, R_{*,j}\rangle| \geq \sqrt{2us^{-2}\log(2n^2/\delta)}\right]$$
$$\leq \delta/n^2 \tag{5}$$

since $u \leq s$. By taking a union bound over all $(i, j) \in [n] \times [n]$ and $i \neq j$, we complete the proof. $\qquad \square$

## D.5 Infinite Norm Bound

**Lemma D.23** (SRHT and AMS). *Let $R \in \mathbb{R}^{b \times n}$ denote a subsample randomized Hadamard transform (Definition D.3) or AMS sketching matrix (Definition D.4). Then for any fixed vector $h \in \mathbb{R}^n$ and any fixed vector $g \in \mathbb{R}^n$, the following properties hold:*

$$\Pr_{R \sim \Pi}\left[|(g^\top R^\top R h) - (g^\top h)| > \frac{\log^{1.5}(n/\delta)}{\sqrt{b}}\|g\|_2\|h\|_2\right] \leq \Theta(\delta).$$

*Proof.* We can rewrite $(g^\top R^\top R h) - (g^\top h)$ as follows:,

$$(g^\top R^\top R h) - (g^\top h) = \sum_{i=1}^{n}\sum_{j\in[n]\backslash i}^{n} g_i h_j \langle R_{*,i}, R_{*,j}\rangle + \sum_{i=1}^{n} g_i h_i(\|R_{*,i}\|_2^2 - 1)$$
$$= \sum_{i=1}^{n}\sum_{j\in[n]\backslash i}^{n} g_i h_j \langle \sigma_i \overline{R}_{*,i}, \sigma_j \overline{R}_{*,j}\rangle.$$

where $\sigma_i$'s are independent Rademacher random variables and $\overline{R}_{*,i} = \sigma_i R_{*,i}, \ \forall i \in [n]$, and the second step follows from $\|R_{*,i}\|_2^2 = 1, \forall i \in [n]$.

We define matrix $A \in \mathbb{R}^{n \times n}$ and $B \in \mathbb{R}^{n \times n}$ as follows:

$$A_{i,j} = g_i h_j \cdot \langle \overline{R}_{*,i}, \overline{R}_{*,j}\rangle, \qquad\qquad \forall i \in [n], j \in [n]$$
$$B_{i,j} = g_i h_j \cdot \max_{i' \neq j'}|\langle \overline{R}_{*,i'}, \overline{R}_{*,j'}\rangle| \qquad\qquad \forall i \in [n], j \in [n]$$

We define $A^\circ \in \mathbb{R}^{n \times n}$ to be the matrix $A \in \mathbb{R}^{n \times n}$ with removing diagonal entries, applying Hasonwright inequality (Lemma B.5), we have

$$\Pr_{\sigma}[|\sigma^\top A^\circ \sigma| \geq \tau] \leq 2 \cdot \exp(-c\min\{\tau^2/\|A^\circ\|_F^2, \tau/\|A^\circ\|\})$$

We can upper bound $\|A^\circ\|$ and $\|A^\circ\|_F$.

$$\|A^\circ\| \leq \|A^\circ\|_F$$
$$\leq \|A\|_F$$
$$\leq \|B\|_F$$
$$= \|g\|_2 \cdot \|h\|_2 \cdot \max_{i \neq j}|\langle \overline{R}_{*,i}, \overline{R}_{*,j}\rangle|$$

$$\leq \|g\|_2 \cdot \|h\|_2 \cdot \max_{i \neq j} |\langle \overline{R}_{*,i}, \overline{R}_{*,j} \rangle|.$$

where the forth step follows from $B$ is rank-1.

For SRHT, note $\overline{R}$ has the same distribution as $R$. By Lemma D.19 (for AMS, we use Lemma D.20) with probability at least $1 - \Theta(\delta)$, we have :

$$\max_{i \neq j} |\langle \overline{R}_{*,i}, \overline{R}_{*,j} \rangle| \leq \frac{\sqrt{\log(n/\delta)}}{\sqrt{b}}.$$

Conditioning on the above event holds.

Choosing $\tau = \|g\|_2 \cdot \|h\|_2 \cdot \log^{1.5}(n/\delta)/\sqrt{b}$, we can show that

$$\Pr\left[\left|(g^\top R^\top Rh) - (g^\top h)\right| \geq \|g\|_2 \cdot \|h\|_2 \frac{\log^{1.5}(n/\delta)}{\sqrt{b}}\right] \leq \Theta(\delta).$$

Thus, we complete the proof. $\qquad\square$

**Lemma D.24** (Random Gaussian). *Let $R \in \mathbb{R}^{b \times n}$ denote a random Gaussian matrix (Definition D.2). Then for any fixed vector $h \in \mathbb{R}^n$ and any fixed vector $g \in \mathbb{R}^n$, the following properties hold:*

$$\Pr_{R \sim \Pi}\left[|(g^\top R^\top Rh) - (g^\top h)| > \frac{\log^{1.5}(n/\delta)}{\sqrt{b}} \|g\|_2 \|h\|_2\right] \leq \Theta(\delta).$$

*Proof.* We follow the same procedure as proving Lemma D.23.

We can rewrite $(g^\top R^\top Rh) - (g^\top h)$ as follows:,

$$
\begin{aligned}
(g^\top R^\top Rh) - (g^\top h) &= \sum_{i=1}^n \sum_{j \in [n] \setminus i} g_i h_j \langle R_{*,i}, R_{*,j} \rangle + \sum_{i=1}^n g_i h_i (\|R_{*,i}\|_2^2 - 1) \\
&= \sum_{i=1}^n \sum_{j \in [n] \setminus i} g_i h_j \langle \sigma_i \overline{R}_{*,i}, \sigma_j \overline{R}_{*,j} \rangle + \sum_{i=1}^n g_i h_i (\|R_{*,i}\|_2^2 - 1). \quad (6)
\end{aligned}
$$

where $\sigma_i$'s are independent Rademacher random variables and $\overline{R}$ has the same distribution as $R$.

To bound the first term $\sum_{i=1}^n \sum_{j \in [n] \setminus i} g_i h_j \langle \sigma_i \overline{R}_{*,i}, \sigma_j \overline{R}_{*,j} \rangle$, we define matrix $A \in \mathbb{R}^{n \times n}$ and $B \in \mathbb{R}^{n \times n}$ as follows:

$$
\begin{aligned}
A_{i,j} &= g_i h_j \cdot \langle \overline{R}_{*,i}, \overline{R}_{*,j} \rangle, && \forall i \in [n], j \in [n] \\
B_{i,j} &= g_i h_j \cdot \max_{i' \neq j'} |\langle \overline{R}_{*,i'}, \overline{R}_{*,j'} \rangle| && \forall i \in [n], j \in [n]
\end{aligned}
$$

We define $A^\circ \in \mathbb{R}^{n \times n}$ to be the matrix $A \in \mathbb{R}^{n \times n}$ with removing diagonal entries, applying Hason-wright inequality (Lemma B.5), we have

$$\Pr_\sigma[|\sigma^\top A^\circ \sigma| \geq \tau] \leq 2 \cdot \exp(-c \min\{\tau^2/\|A^\circ\|_F^2, \tau/\|A^\circ\|\})$$

We can upper bound $\|A^\circ\|$ and $\|A^\circ\|_F$.

$$
\begin{aligned}
\|A^\circ\| &\leq \|A^\circ\|_F \\
&\leq \|A\|_F \\
&\leq \|B\|_F \\
&= \|g\|_2 \cdot \|h\|_2 \cdot \max_{i \neq j} |\langle \overline{R}_{*,i}, \overline{R}_{*,j} \rangle| \\
&\leq \|g\|_2 \cdot \|h\|_2 \cdot \max_{i \neq j} |\langle \overline{R}_{*,i}, \overline{R}_{*,j} \rangle|.
\end{aligned}
$$

where the forth step follows from $B$ is rank-1.

Using Lemma D.18 with probability at least $1 - \Theta(\delta)$, we have :

$$\max_{i \neq j} |\langle \overline{R}_{*,i}, \overline{R}_{*,j} \rangle| \leq \frac{\sqrt{\log(n/\delta)}}{\sqrt{b}}.$$

Conditioning on the above event holds.

Choosing $\tau = \|g\|_2 \cdot \|h\|_2 \cdot \log^{1.5}(n/\delta)/\sqrt{b}$, we can show that

$$\Pr\left[ \Big| \sum_{i=1}^{n} \sum_{j \in [n] \setminus i} g_i h_j \langle \sigma_i \overline{R}_{*,i}, \sigma_j \overline{R}_{*,j} \rangle \Big| \geq \|g\|_2 \cdot \|h\|_2 \frac{\log^{1.5}(n/\delta)}{\sqrt{b}} \right] \leq \Theta(\delta). \quad (7)$$

To bound the second term $\sum_{i=1}^{n} g_i h_i (\|R_{*,i}\|_2^2 - 1)$, note that $b\|R_{*,i}\|_2^2 \sim \chi_b^2$ for every $i \in [n]$. Applying Lemma B.6, we have

$$\Pr\left[ \Big| \|R_{*,i}\|_2^2 - 1 \Big| \geq \frac{c\sqrt{\log(n/\delta)}}{\sqrt{b}} \right] \leq \delta/n.$$

which implies

$$\Pr\left[ \sum_{i=1}^{n} g_i h_i \Big| \|R_{*,i}\|_2^2 - 1 \Big| \geq \|g\|_2 \|h\|_2 \frac{c\sqrt{\log(n/\delta)}}{\sqrt{b}} \right] \leq \Theta(\delta). \quad (8)$$

Plugging the bounds Eq. (7) and (8) back to Eq. (6), we complete the proof. □

**Lemma D.25** (Count-sketch). *Let $R \in \mathbb{R}^{b \times n}$ denote a count-sketch matrix (Definition D.5). Then for any fixed vector $h \in \mathbb{R}^n$ and any fixed vector $g \in \mathbb{R}^n$, the following properties hold:*

$$\Pr_{R \sim \Pi}\left[ |(g^\top R^\top R h) - (g^\top h)| \geq \log(1/\delta) \|g\|_2 \|h\|_2 \right] \leq \Theta(\delta).$$

*Proof.* We follow the identical procedure as proving Lemma D.23 to apply Hason-wright inequality (Lemma B.5).

Then note Lemma D.21 shows

$$\max_{i \neq j} |\langle \overline{R}_{*,i}, \overline{R}_{*,j} \rangle| \leq 1$$

Thus, choosing $\tau = c\|g\|_2 \cdot \|h\|_2 \cdot \log(1/\delta)$, we can show that

$$\Pr\left[ |(g^\top R^\top R h) - (g^\top h)| \geq c\|g\|_2 \cdot \|h\|_2 \log(1/\delta) \right] \leq \delta.$$

which completes the proof. □

**Lemma D.26** (Count-sketch 2). *Let $R \in \mathbb{R}^{b \times n}$ denote a count-sketch matrix (Definition D.5). Then for any fixed vector $h \in \mathbb{R}^n$ and any fixed vector $g \in \mathbb{R}^n$, the following properties hold:*

$$\Pr_{R \sim \Pi}\left[ |(g^\top R^\top R h) - (g^\top h)| \geq \frac{1}{\sqrt{b\delta}} \|g\|_2 \|h\|_2 \right] \leq \Theta(\delta).$$

*Proof.* It is known that a count-sketch matrix with $b = \epsilon^{-2}\delta^{-1}$ rows satisfies the $(\epsilon, \delta, 2)$-JL moment property (see e.g. Theorem 14 of Woodruff (2014)). Using Markov's inequality, $(\epsilon, \delta, 2)$-JL moment property implies

$$\Pr_{R \sim \Pi}\left[ |(g^\top R^\top R h) - (g^\top h)| \geq \epsilon \|g\|_2 \|h\|_2 \right] \leq \Theta(\delta),$$

where $\epsilon = \frac{1}{\sqrt{b\delta}}$. □

**Remark D.27.** *In LP solver, we need $\delta = 1/\operatorname{poly}(n)$, thus Lemma D.25 is stronger than Lemma D.26.*

**Lemma D.28** (Sparse embedding). *Let $R \in \mathbb{R}^{b \times n}$ denote a sparse-embedding matrix (Definition D.6 and D.7). Then for any fixed vector $h \in \mathbb{R}^n$ and any fixed vector $g \in \mathbb{R}^n$, the following properties hold:*

$$3. \Pr_{R \sim \Pi}\left[|(g^\top R^\top R h) - (g^\top h)| > \frac{\log^{1.5}(n/\delta)}{\sqrt{s}}\|g\|_2\|h\|_2\right] \leq \Theta(\delta).$$

*Proof.* We follow the identical procedure as proving Lemma D.23 to apply Hason-wright inequality (Lemma B.5).

Then note Lemma D.22 shows with probability at least $1 - \delta$ we have

$$\max_{i \neq j}|\langle \overline{R}_{*,i}, \overline{R}_{*,j}\rangle| \leq \frac{c\sqrt{\log(n/\delta)}}{\sqrt{s}}.$$

Conditioning on the above event holds, choosing $\tau = c'\|g\|_2 \cdot \|h\|_2 \cdot \log^{1.5}(1/\delta)$, we can show that

$$\Pr\left[|(g^\top R^\top R h) - (g^\top h)| \geq \frac{c'\log^{1.5}(n/\delta)}{\sqrt{s}}\|g\|_2 \cdot \|h\|_2\right] \leq \Theta(\delta).$$

Thus, we complete the proof. $\qquad\square$

**Lemma D.29** (Uniform sampling). *Let $R \in \mathbb{R}^{b \times n}$ denote a uniform sampling matrix (Definition D.8). Then for any fixed vector $h \in \mathbb{R}^n$ and any fixed vector $g \in \mathbb{R}^n$, the following properties hold:*

$$3. |(g^\top R^\top R h) - (g^\top h)| \leq (1 + \frac{n}{b})\|g\|_2\|h\|_2$$

*where $I \subset [n]$ be the subset of indexes chosen by the uniform sampling matrix.*

*Proof.* We can rewrite $(g^\top R^\top R h) - (g^\top h)$ as follows:,

$$(g^\top R^\top R h) - (g^\top h) = \sum_{i=1}^n \sum_{j \in [n] \setminus i} g_i h_j \langle R_{*,i}, R_{*,j}\rangle + \sum_{i=1}^n g_i h_i(\|R_{*,i}\|_2^2 - 1)$$

$$= \frac{n}{b}\sum_{i \in I} g_i h_i - \sum_{i=1}^n g_i h_i.$$

where the second step follows from the uniform sampling matrix has only one nonzero entry in each row.

Let $I \subset [n]$ be the subset chosen by the uniform sampling matrix, then $\|R_{*,i}\|_2^2 = n/b$ for $i \in I$ and $\|R_{*,i}\|_2^2 = 0$ for $i \in [n] \setminus I$. So we have

$$|(g^\top R^\top R h) - (g^\top h)| = \left|\sum_{i \in I} g_i h_i(\frac{n}{b} - 1) - \sum_{i \in [n] \setminus I} g_i h_i\right|$$

$$\leq (1 + \frac{n}{b})\|g\|_2\|h\|_2.$$

$$\square$$

# E    ANALYSIS OF CONVERGENCE: SINGLE-STEP SCHEME

## E.1    PRELIMINARY

Throughout the proof of convergence, we will use $\mathcal{F}_t$ to denote the sequence $w_{t-1}, w_{t-2}, \ldots, w_0$. Also, we use $\eta$ as a shorthand for $\eta_{\text{global}} \cdot \eta_{\text{local}}$.

### E.2 STRONGLY-CONVEX $f$ CONVERGENCE ANALYSIS

**Theorem E.1.** *Let* $f : \mathbb{R}^d \to \mathbb{R}$ *satisfying Assumption 3.1 with* $\mu > 0$. *Let* $w^* \in \mathbb{R}^d$ *be the optimal solution to* $f$ *and assume* sk/desk *functions satisfying Theorem 4.2. Suppose* $\eta := \eta_{\text{global}} \cdot \eta_{\text{local}}$ *has the property that* $\eta \leq \frac{1}{(1+\alpha)L}$, *then*

$$\mathbb{E}[f(w^{t+1})] - f(w^*) \leq (1 - \mu\eta)^t \cdot (f(w^0) - f(w^*))$$

*Proof.* We shall first bound $f(w^{t+1}) - f(w^t)$:

$$
\begin{aligned}
f(w^{t+1}) - f(w^t) \leq & \langle w^{t+1} - w^t, \nabla f(w^t) \rangle + \frac{L}{2}\|w^{t+1} - w^t\|_2^2 \\
= & \langle \text{desk}_t(\Delta\widetilde{w}^t), \nabla f(w^t) \rangle + \frac{L}{2}\|\text{desk}_t(\Delta\widetilde{w}^t)\|_2^2 \\
= & -\langle \eta_{\text{global}} \cdot \text{desk}_t(\frac{1}{N}\sum_{c=1}^{N}\text{sk}_t(\eta_{\text{local}} \cdot \nabla f_c(w^t))), \nabla f(w^t) \rangle \\
& + \frac{L}{2}\|\eta_{\text{global}} \cdot \text{desk}_t(\frac{1}{N}\sum_{c=1}^{N}\text{sk}_t(\eta_{\text{local}} \cdot \nabla f_c(w^t)))\|_2^2 \\
= & -\eta_{\text{global}} \cdot \eta_{\text{local}} \cdot \langle \text{desk}_t(\text{sk}_t(\nabla f(w^t))), \nabla f(w^t) \rangle \\
& + (\eta_{\text{global}} \cdot \eta_{\text{local}})^2 \cdot \|\text{desk}_t(\text{sk}_t(\nabla f(w^t)))\|_2^2
\end{aligned}
$$

where the first step uses the $L$-smoothness condition of $f$, and the last step uses the linearity property of sk/desk functions.

Taking expectation over iteration $t$ conditioning on $\mathcal{F}_t$ and note that only $w^{t+1}$ depends on randomness at $t$, we get

$$
\begin{aligned}
& \mathbb{E}[f(w^{t+1}) - f(w^t) \mid \mathcal{F}_t] \\
\leq & -\eta \cdot \langle \mathbb{E}[\text{desk}_t(\text{sk}_t(\nabla f(w^t))) \mid \mathcal{F}_t], \nabla f(w^t) \rangle + \frac{L\eta^2}{2}\mathbb{E}[\|\text{desk}_t(\text{sk}_t(\nabla f(w^t)))\|_2^2 \mid \mathcal{F}_t] \\
\leq & -\eta \cdot \langle \nabla f(w^t), \nabla f(w^t) \rangle + \frac{L\eta^2}{2}(1 + \alpha) \cdot \|\nabla f(w^t)\|_2^2 \\
\leq & -\frac{\eta}{2} \cdot \|\nabla f(w^t)\|_2^2 \\
\leq & -\mu\eta \cdot (f(w^t) - f(w^*))
\end{aligned} \tag{9}
$$

where the second step comes from the fact that $\text{desk}_t(\text{sk}_t(h))$ is an unbiased estimator for any fixed $h \in \mathbb{R}^d$ and the bound on its variance, the third step comes from $\eta \leq \frac{1}{(1+\alpha)L}$, and the last step comes from Fact C.5.

Upon rearranging and subtracting both sides by $f(w^*)$, we get

$$\mathbb{E}[f(w^{t+1})] - f(w^*) \mid \mathcal{F}_t \leq (1 - \mu\eta) \cdot (f(w^t) - f(w^*)) \tag{10}$$

Note that if we apply expectation over $\mathcal{F}_t$ on both sides of Eq. (10) we can get

$$\mathbb{E}[f(w^{t+1})] - f(w^*) \leq (1 - \mu\eta) \cdot (\mathbb{E}[f(w^t)] - f(w^*)) \tag{11}$$

Notice since $1 - \mu\eta \leq 1$, this is a contraction map, if we iterate this recurrence relation, we will finally get

$$\mathbb{E}[f(w^{t+1}) - f(w^*)] \leq (1 - \mu\eta)^t \cdot (f(w^0) - f(w^*)). \tag{12}$$

$\square$

### E.3 CONVEX $f$ CONVERGENCE ANALYSIS

Assume $f$ is a convex function, we obtain a convergence bound in terms of the average of all parameters.

**Theorem E.2.** *Let $f : \mathbb{R}^d \to \mathbb{R}$ satisfying Assumption 3.1 with $\mu = 0$. Suppose* sk/desk *functions satisfying Theorem 4.2. If $\eta := \eta_{\text{global}} \cdot \eta_{\text{local}} \leq \frac{1}{2(1+\alpha)L}$, then*

$$\mathbb{E}[f(\overline{w}^T) - f(w^*)] \leq \frac{\mathbb{E}[\|w^0 - w^*\|_2^2]}{\eta \cdot (T+1)}$$

*where $\overline{w}^T := \frac{1}{T+1} \sum_{t=0}^{T} w^t$ and $w^* \in \mathbb{R}^d$ is the optimal solution.*

*Proof.* We shall first compute the gap between $w^{t+1}$ and $w^*$:

$$
\begin{aligned}
&\|w^{t+1} - w^*\|_2^2 \\
&= \|w^t - \mathsf{desk}_t(\Delta \widetilde{w}^t) - w^*\|_2^2 \\
&= \|w^t - \eta \cdot \mathsf{desk}_t(\mathsf{sk}_t(\nabla f(w^t))) - w^*\|_2^2 \\
&= \|w^t - w^*\|_2^2 + \eta^2 \cdot \|\mathsf{desk}_t(\mathsf{sk}_t(\nabla f(w^t)))\|_2^2 - 2\eta \cdot \langle w^t - w^*, \mathsf{desk}_t(\mathsf{sk}_t(\nabla f(w^t)))\rangle \quad (13)
\end{aligned}
$$

By unbiasedness of $\mathsf{desk}_t \circ \mathsf{sk}_t$, we have

$$\mathbb{E}[\langle w^t - w^*, \mathsf{desk}_t(\mathsf{sk}_t(\nabla f(w^t)))\rangle \mid \mathcal{F}_t] = \mathbb{E}[\langle w^t - w^*, \nabla f(w^t)\rangle \mid \mathcal{F}_t] \quad (14)$$

Taking total expectation of Eq. (13) and plug in Eq. (14), we get

$$
\begin{aligned}
&\mathbb{E}[\|w^{t+1} - w^*\|_2^2 \mid \mathcal{F}_t] \\
&= \mathbb{E}[\|w^t - w^*\|_2^2 \mid \mathcal{F}_t] + \eta^2 \cdot \mathbb{E}[\|\mathsf{desk}_t(\mathsf{sk}_t(\nabla f(w^t)))\|_2^2 \mid \mathcal{F}_t] - 2\eta \cdot \mathbb{E}[\langle w^t - w^*, \nabla f(w^t)\rangle \mid \mathcal{F}_t] \\
&\leq \mathbb{E}[\|w^t - w^*\|_2^2 \mid \mathcal{F}_t] + \eta^2 \cdot (1+\alpha) \cdot \mathbb{E}[\|\nabla f(w^t)\|_2^2 \mid \mathcal{F}_t] + 2\eta \cdot \mathbb{E}[\langle w^* - w^t, \nabla f(w^t)\rangle \mid \mathcal{F}_t] \\
&\leq \mathbb{E}[\|w^t - w^*\|_2^2 \mid \mathcal{F}_t] + \eta^2 \cdot (1+\alpha) \cdot \mathbb{E}[\|\nabla f(w^t)\|_2^2 \mid \mathcal{F}_t] + 2\eta \cdot \mathbb{E}[f(w^*) - f(w^t) \mid \mathcal{F}_t] \quad (15)
\end{aligned}
$$

where the second step follows from the variance of $\mathsf{desk}_t \circ \mathsf{sk}_t$, and the last step follows from the convexity of $f$. Taking the expectation over $\mathcal{F}_t$ and re-organizing the above equation, we can get

$$
\begin{aligned}
2\eta \cdot \mathbb{E}[f(w^t) - f(w^*)] &\leq \mathbb{E}[\|w^t - w^*\|_2^2] - \mathbb{E}[\|w^{t+1} - w^*\|_2^2] + \eta^2 \cdot (1+\alpha) \cdot \mathbb{E}[\|\nabla f(w^t)\|_2^2] \\
&\leq \mathbb{E}[\|w^t - w^*\|_2^2] - \mathbb{E}[\|w^{t+1} - w^*\|_2^2] + \eta^2 \cdot (1+\alpha) \cdot 2L \cdot \mathbb{E}[f(w^t) - f(w^*)]
\end{aligned}
$$

where the second step follows from the convexity and $L$-smoothness of $f$. Rearrange the above inequality, we have

$$(2\eta - \eta^2 \cdot (1+\alpha) \cdot 2L) \cdot \mathbb{E}[f(w^t) - f(w^*)] \leq \mathbb{E}[\|w^t - w^*\|_2^2] - \mathbb{E}[\|w^{t+1} - w^*\|_2^2]$$

Note $\eta \leq \frac{1}{2(1+\alpha)L}$, we have

$$\eta \cdot \mathbb{E}[f(w^t) - f(w^*)] \leq \mathbb{E}[\|w^t - w^*\|_2^2] - \mathbb{E}[\|w^{t+1} - w^*\|_2^2]$$

Sum over all $T$ iterations, we arrive at

$$\eta \cdot \sum_{t=0}^{T} \mathbb{E}[f(w^t) - f(w^*)] \leq \mathbb{E}[\|w^0 - w^*\|_2^2] - \mathbb{E}[\|w^{T+1} - w^*\|_2^2] \leq \mathbb{E}[\|w^0 - w^*\|_2^2] \quad (16)$$

Let $\overline{w}^T = \frac{1}{T+1} \sum_{t=0}^{T} w^t$ denote the average of parameters across iterations, then by convexity of $f$, we conclude:

$$\mathbb{E}[f(\overline{w}^T) - f(w^*)] \leq \frac{\mathbb{E}[\|w^0 - w^*\|_2^2]}{\eta \cdot (T+1)}$$

$\square$

### E.4 NON-CONVEX $f$ CONVERGENCE ANALYSIS

Next, we prove a version when $f$ is not even a convex function, due to loss of convexity, we can no longer bound the gap between $\mathbb{E}[f(w^t)]$ and $f(w^*)$, but we can instead bound the minimum (or average) expected gradient.

**Theorem E.3.** *Let $f : \mathbb{R}^d \to \mathbb{R}$ be an L-smooth function (Def. C.1) and $\mathsf{sk/desk}$ functions satisfying Theorem 4.2, let $w^* \in \mathbb{R}^d$ be the optimal solution to $f$. Suppose $\eta := \eta_{\text{local}} \cdot \eta_{\text{global}} \leq \frac{1}{(1+\alpha)L}$, then*

$$\min_{t \in [T]} \mathbb{E}[\|\nabla f(w^t)\|_2^2] \leq \frac{2}{\eta(T+1)}(\mathbb{E}[f(w^0)] - f(w^*))$$

*Proof.* Note that the only place we used strongly-convex assumption in the proof of Theorem E.1 is Eq. (9), so by the same analysis, we can get

$$\mathbb{E}[f(w^{t+1}) - f(w^t) \mid \mathcal{F}_t] \leq -\frac{\eta}{2} \cdot \|\nabla f(w^t)\|_2^2$$

Rearranging and taking total expectation over $\mathcal{F}_t$, we get

$$\mathbb{E}[\|\nabla f(w^t)\|_2^2] \leq \frac{2}{\eta}(\mathbb{E}[f(w^t)] - \mathbb{E}[f(w^{t+1})])$$

Averaging over all $T$ iterations, we get

$$\frac{1}{T+1}\sum_{t=0}^{T}\mathbb{E}[\|\nabla f(w^t)\|_2^2] \leq \frac{2}{\eta(T+1)}\sum_{t=0}^{T}(\mathbb{E}[f(w^t)] - \mathbb{E}[f(w^{t+1})])$$

$$= \frac{2}{\eta(T+1)}(\mathbb{E}[f(w^0)] - \mathbb{E}[f(w^T)])$$

$$\leq \frac{2}{\eta(T+1)}(\mathbb{E}[f(w^0)] - f(w^*))$$

This implies our final result:

$$\min_{t \in [T]} \mathbb{E}[\|\nabla f(w^t)\|_2^2] \leq \frac{2}{\eta(T+1)}(\mathbb{E}[f(w^0)] - f(w^*))$$

$\square$

**Remark E.4.** *Notice due to the structure of $\mathsf{sk/desk}$ functions, i.e., their variance is bounded in terms of true gradient, the convergence rate depends completely on the term $\frac{1}{(1+\alpha)L}$. If it's a constant, then we essentially recover a convergence rate of gradient descent. On the other hand, if $\frac{1}{(1+\alpha)L} \leq \frac{1}{\sqrt{T}}$, then we get a similar convergence rate as SGD. One clear advantage of our $\mathsf{sk/desk}$ functions is they don't introduce extra noise term as in SGD, since we can choose appropriate step size to absorb the variance term.*

# F  $k$-STEP CONVEX & STRONGLY-CONVEX $f_c$ ANALYSIS

## F.1  PRELIMINARY

In this section, we assume each $f_c$ satisfies Assumption 3.1 and $\eta_{\text{global}} = 1$. For notation simplicity, we also denote $u_c^{t,-1} = u_c^{t-1,K-1}$ for $t \geq 2$.

**Definition F.1.** *Let $(t, k) \in \{1, \cdots, T+1\} \times \{-1, 0, 1, \cdots, K-1\}$, we define the following terms for iteration $(t, k)$:*

$$\overline{u}^{t,k} := \frac{1}{N}\sum_{c=1}^{N} u_c^{t,k}, \quad r^{t,k} := \overline{u}^{t,k} - w^*$$

*to be the average of local parameters and its distance to the optimal solution,*

$$g_c^{t,k} := \nabla f_c(u_c^{t,k}), \quad \overline{g}^{t,k} := \frac{1}{N}\sum_{c=1}^{N} \nabla f_c(u_c^{t,k})$$

*to be the local gradient and its average,*

$$V^{t,k} := \frac{1}{N}\sum_{c=1}^{N} \|u_c^{t,k} - \overline{u}^{t,k}\|_2^2$$

*to be the variances of local updates,*

$$\sigma^2 = \frac{1}{N} \sum_{c=1}^{N} \|\nabla f_c(w^*)\|^2$$

*to be a finite constant that characterize the heterogeneity of local objectives.*

*We also define the following indicator function: let $l \in \mathbb{R}$, then we define $1_{\{x=l\}}$ to be*

$$1_{\{x=l\}} = \begin{cases} 1 & \text{if } x = l, \\ 0 & \text{otherwise.} \end{cases}$$

## F.2 UNIFYING THE UPDATE RULE OF ALGORITHM 1

**Lemma F.2.** *We have the following facts for $u_c^{t,k}$ and $\widetilde{u}^{t,k}$:*

$u_c^{t,0} = \overline{u}^{t,0}$

$u_c^{t,k} = u_c^{t,k-1} - \eta_{\text{local}} \cdot g_c^{t,k-1}, \ \forall k \geq 1$

$\overline{u}^{t,k} = \overline{u}^{t,k-1} - \eta_{\text{local}} \cdot \overline{g}^{t,k-1} + 1_{\{k=0\}} \cdot \eta_{\text{local}} \cdot (I_d - \mathsf{desk}_t \circ \mathsf{sk}_t)(\sum_{i=0}^{K-1} \overline{g}^{t-1,i}), \ \forall (t,k) \neq (1,0)$

*where $I_d : \mathbb{R}^d \to \mathbb{R}^d$ is the identity function.*

*Proof.* First two equation directly follows from the update rule of Algorithm 1.
For $k = 1, 2, \cdots, K-1$, taking the average of the second equation we obtain:

$$\overline{u}^{t,k} = \overline{u}^{t,k-1} - \eta_{\text{local}} \cdot \overline{g}^{t,k-1}$$

For $k = 0$ and $t \geq 2$, we have

$$\overline{u}^{t,0} = \overline{u}^{t-1,0} - \eta_{\text{local}} \cdot \mathsf{desk}_t(\mathsf{sk}_t(\sum_{i=0}^{K-1} \overline{g}^{t-1,i}))$$

$$= \overline{u}^{t-1,0} - \eta_{\text{local}} \sum_{i=0}^{K-1} \overline{g}^{t-1,i} + \eta_{\text{local}} \sum_{i=0}^{K-1} \overline{g}^{t-1,i} - \eta_{\text{local}} \cdot \mathsf{desk}_t(\mathsf{sk}_t(\sum_{i=0}^{K-1} \overline{g}^{t-1,i}))$$

$$= \overline{u}^{t-1,K-1} - \eta_{\text{local}} \cdot \overline{g}^{t-1,K-1} + \eta_{\text{local}} \cdot (I_d - \mathsf{desk}_t \circ \mathsf{sk}_t)(\sum_{i=0}^{K-1} \overline{g}^{t-1,i})$$

Combining above results together, we prove the third equation. $\square$

## F.3 UPPER BOUNDING $\|\overline{g}^{t,k}\|_2^2$

**Lemma F.3.** *Suppose for any $c \in [N]$, $f_c : \mathbb{R}^d \to \mathbb{R}$ is convex and $L$-smooth. Then*

$$\|\overline{g}^{t,k}\|_2^2 \leq 2L^2 V^{t,k} + 4L(f(\overline{u}^{t,k}) - f(w^*))$$

*Proof.* By triangle inequality and Cauchy-Schwartz inequality, we have

$$\|\overline{g}^{t,k}\|_2^2 = \|\overline{g}^{t,k} - \nabla f(\overline{u}^{t,k}) + \nabla f(\overline{u}^{t,k})\|_2^2$$
$$\leq 2\|g^{t,k} - \nabla f(\overline{u}^{t,k})\|_2^2 + 2\|\nabla f(\overline{u}^{t,k})\|_2^2$$

where the first term can be bounded as

$$\|\overline{g}^{t,k} - \nabla f(\overline{u}^{t,k})\|_2^2 = \|\frac{1}{N} \sum_{c=1}^{N} \nabla f_c(u_c^{t,k}) - \frac{1}{N} \sum_{c=1}^{N} \nabla f_c(\overline{u}^{t,k})\|_2^2$$

$$\leq \frac{1}{N} \sum_{c=1}^{N} \|\nabla f_c(u_c^{t,k}) - f_c(\overline{u}^{t,k})\|_2^2$$

$$\leq \frac{L^2}{N} \sum_{c=1}^{N} \|u_c^{t,k} - \overline{u}^{t,k}\|_2^2$$

and the second term can be bounded as follows:

$$\|\nabla f(\overline{u}^{t,k})\|_2^2 = \|\nabla f(\overline{u}^{t,k}) - \nabla f(w^*)\|_2^2$$
$$\leq 2L(f(\overline{u}^{t,k}) - f(w^*))$$

where the last step follows from that $f$ is $L$-smooth and Fact C.4.

Combining bounds on these two terms, we get

$$\|\overline{g}^{t,k}\|_2^2 \leq \frac{2L^2}{N} \sum_{c=1}^{N} \|u_c^{t,k} - \overline{u}^{t,k}\|_2^2 + 2L^2\|\overline{u}^{t,k} - w^*\|_2^2$$
$$\leq 2L^2 V^{t,k} + 4L(f(\overline{u}^{t,k}) - f(w^*))$$

$\square$

## F.4   LOWER BOUNDING $\langle \overline{u}^{t,k} - w^*, \overline{g}^{t,k} \rangle$

**Lemma F.4.** *Suppose each $f_c$ satisfies Assumption 3.1 with $\mu \geq 0$, then*

$$\langle \overline{u}^{t,k} - w^*, \overline{g}^{t,k} \rangle \geq f(\overline{u}^{t,k}) - f(w^*) - \frac{L}{2} V^{t,k} + \frac{\mu}{2}\|\overline{u}^{t,k} - w^*\|_2^2$$

*Proof.* We will provide a lower bound on this inner product:

$$\langle \overline{u}^{t,k} - w^*, \overline{g}^{t,k} \rangle = \frac{1}{N} \sum_{c=1}^{N} \langle \overline{u}^{t,k} - w^*, \nabla f_c(u_c^{t,k}) \rangle$$

It suffices to consider each term separately:

$$\langle \overline{u}^{t,k} - w^*, \nabla f_c(u_c^{t,k}) \rangle = \langle \overline{u}^{t,k} - u_c^{t,k} + u_c^{t,k} - w^*, \nabla f_c(u_c^{t,k}) \rangle$$
$$= \langle \overline{u}^{t,k} - u_c^{t,k}, \nabla f_c(u_c^{t,k}) \rangle + \langle u_c^{t,k} - w^*, \nabla f_c(u_c^{t,k}) \rangle$$

The first term can be lower bounded via $L$-smoothness:

$$\langle \overline{u}^{t,k} - u_c^{t,k}, \nabla f_c(u_c^{t,k}) \rangle \geq f_c(\overline{u}^{t,k}) - f_c(u_c^{t,k}) - \frac{L}{2}\|\overline{u}^{t,k} - u_c^{t,k}\|_2^2$$

The second term can be lower bounded via convexity:

$$\langle u_c^{t,k} - w^*, \nabla f_c(u_c^{t,k}) \rangle \geq f_c(u_c^{t,k}) - f_c(w^*) + \frac{\mu}{2}\|u_c^{t,k} - w^*\|_2^2$$

Combining these two bounds and average them, we get a lower bound:

$$\langle \overline{u}^{t,k} - w^*, g^{t,k} \rangle \geq \frac{1}{N} \sum_{c=1}^{N} (f_c(\overline{u}^{t,k}) - f_c(w^*) - \frac{L}{2}\|\overline{u}^{t,k} - u_c^{t,k}\|_2^2 + \frac{\mu}{2}\|u_c^{t,k} - w^*\|_2^2)$$

$$\geq \frac{1}{N} \sum_{c=1}^{N} (f_c(\overline{u}^{t,k}) - f_c(w^*)) - \frac{L}{2}V^{t,k} + \frac{\mu}{2}\|\overline{u}^{t,k} - w^*\|_2^2$$

$$= f(\overline{u}^{t,k}) - f(w^*) - \frac{L}{2}V^{t,k} + \frac{\mu}{2}\|\overline{u}^{t,k} - w^*\|_2^2$$

$\square$

### F.5 Upper bounding variance within $K$ local steps

**Lemma F.5.** *Suppose each $f_c$ is convex and $L$-smooth. Assume $\eta_{\text{local}} \leq \frac{1}{8LK}$. Then for any $t \geq 0$,*

$$\sum_{k=0}^{K-1} V^{t,k} \leq 8\eta_{\text{local}}^2 LK^2 \sum_{k=0}^{K-1}(f(\overline{u}^{t,k}) - f(w^*)) + 4\eta_{\text{local}}^2 K^3 \sigma^2$$

*Proof.* By Lemma F.2, we know $V^{t,0} = 0$ for any $t \geq 0$. Consider $k \in \{1, 2, \cdots, K-1\}$, we have

$$V^{t,k} = \frac{1}{N}\sum_{c=1}^{N}\|u_c^{t,k} - \overline{u}^{t,k}\|_2^2$$

$$= \frac{1}{N}\sum_{c=1}^{N}\|u_c^{t,0} - \sum_{i=0}^{k-1}\eta_{\text{local}} \cdot g_c^{t,i} - \overline{u}^{t,0} + \sum_{i=0}^{k-1}\eta_{\text{local}} \cdot \overline{g}^{t,i}\|_2^2$$

$$= \frac{\eta_{\text{local}}^2}{N}\sum_{c=1}^{N}\|\sum_{i=0}^{k-1}(\overline{g}^{t,i} - g_c^{t,i})\|_2^2$$

$$\leq \frac{\eta_{\text{local}}^2 k}{N}\sum_{c=1}^{N}\sum_{i=0}^{k-1}\|\overline{g}^{t,i} - g_c^{t,i}\|_2^2$$

$$\leq \frac{\eta_{\text{local}}^2 K}{N}\sum_{c=1}^{N}\sum_{i=0}^{k-1}\|g_c^{t,i}\|_2^2 \tag{17}$$

where the second step follows from Lemma F.2, the last step follows from $\overline{g}^{t,i}$ being the average of $g_c^{t,i}$. By Cauchy-Schwartz inequality, we further have:

$$\|g_c^{t,i}\|_2^2 \leq 3\|g_c^{t,i} - \nabla f_c(\overline{u}^{t,i})\|_2^2 + 3\|\nabla f_c(\overline{u}^{t,i}) - \nabla f_c(w^*)\|_2^2 + 3\|\nabla f_c(w^*)\|_2^2$$
$$\leq 3L^2\|u_c^{t,i} - \overline{u}^{t,i}\|_2^2 + 6L(f_c(\overline{u}^{t,i}) - f_c(w^*) + \langle w^* - \overline{u}^{t,0}, \nabla f_c(w^*)\rangle) + 3\|\nabla f_c(w^*)\|_2^2.$$

where the last step follows from applying $L$-smoothness to the first and second term.

Averaging with respect to $c$,

$$\frac{1}{N}\sum_{c=1}^{N}\|g_c^{t,i}\|_2^2 \leq 3L^2 V^{t,i} + 6L(f(\overline{u}^{t,i}) - f(w^*)) + 3\sigma^2.$$

Note that the inner product term vanishes since $\frac{1}{N}\sum_{c=1}^{N}\nabla f_c(w^*) = \nabla f(w^*) = 0$.

Plugging back to Eq. (17), we obtain

$$V^{t,k} \leq \frac{\eta_{\text{local}}^2 K}{N}\sum_{c=1}^{N}\sum_{i=0}^{k-1}\|g_c^{t,i}\|_2^2$$

$$\leq \eta_{\text{local}}^2 K\sum_{i=0}^{k-1}(3L^2 V^{t,i} + 6L(f(\overline{u}^{t,i}) - f(w^*)) + 3\sigma^2).$$

Summing up above inequality as $k$ varies from 0 to $K-1$,

$$\sum_{k=0}^{K-1} V^{t,k} \leq \eta_{\text{local}}^2 K\sum_{k=0}^{K-1}\sum_{i=0}^{k-1}(3L^2 V^{t,i} + 6L(f(\overline{u}^{t,i}) - f(w^*)) + 3\sigma^2)$$

$$\leq \eta_{\text{local}}^2 K\sum_{k=0}^{K-1}\sum_{i=0}^{K-1}(3L^2 V^{t,i} + 6L(f(\overline{u}^{t,i}) - f(w^*)) + 3\sigma^2)$$

$$= 3\eta_{\text{local}}^2 L^2 K^2\sum_{i=0}^{K-1} V^{t,i} + 6\eta_{\text{local}}^2 LK^2\sum_{i=0}^{K-1}(f(\overline{u}^{t,i}) - f(w^*)) + 3\eta_{\text{local}}^2 K^3\sigma^2$$

Rearranging terms we obtain:

$$(1 - 3\eta_{\text{local}}^2 L^2 K^2) \sum_{k=0}^{K-1} V^{t,k} \leq 6\eta_{\text{local}}^2 LK^2 \sum_{i=0}^{K-1} (f(\overline{u}^{t,i}) - f(w^*)) + 3\eta_{\text{local}}^2 K^3 \sigma^2$$

Since $\eta_{\text{local}} \leq \frac{1}{8LK}$, we have $1 - 3\eta_{\text{local}}^2 L^2 K^2 \geq \frac{3}{4}$, implying

$$\sum_{k=0}^{K-1} V^{t,k} \leq 8\eta_{\text{local}}^2 LK^2 \sum_{i=0}^{K-1} (f(\overline{u}^{t,i}) - f(w^*)) + 4\eta_{\text{local}}^2 K^3 \sigma^2$$

□

## F.6 BOUNDING THE EXPECTED GAP BETWEEN $\overline{u}^{t,k}$ AND $w^*$

**Lemma F.6.** *Suppose each $f_c$ satisfies Assumption 3.1 with $\mu \geq 0$. If sk/desk satisfying Theorem 4.2 and $\eta_{\text{local}} \leq \frac{1}{4L}$, then for any $(t,k) \neq (1,0)$, we have*

$$\mathbb{E}[\|\overline{u}^{t,k} - w^*\|_2^2] \leq (1 - \mu\eta_{\text{local}})\,\mathbb{E}[\|\overline{u}^{t,k-1} - w^*\|_2^2] + \frac{3}{2}\eta_{\text{local}} L\,\mathbb{E}[V^{t,k-1}] - \eta_{\text{local}}\,\mathbb{E}[f(\overline{u}^{t,k-1}) - f(w^*)]$$

$$+ 1_{\{k=0\}}\eta_{\text{local}}^2 \alpha K \Big(2L^2 \sum_{i=0}^{K-1} \mathbb{E}[V^{t-1,i}] + 4L \sum_{i=0}^{K-1} \mathbb{E}[f(\overline{u}^{t-1,i}) - f(w^*)]\Big)$$

*Proof.* By Lemma F.2, we have for any $(t,k) \neq (1,0)$,

$$\overline{u}^{t,k} = \overline{u}^{t,k-1} - \eta_{\text{local}} \cdot \overline{g}^{t,k-1} + 1_{\{k=0\}} \cdot \eta_{\text{local}} \cdot (I_d - \text{desk}_t \circ \text{sk}_t)(\sum_{i=0}^{K-1} \overline{g}^{t-1,i})$$

Therefore, denoting $h^t := (I_d - \text{desk}_t \circ \text{sk}_t)(\sum_{i=0}^{K-1} \overline{g}^{t-1,i})$, we have

$$\|\overline{u}^{t,k} - w^*\|_2^2 = \|\overline{u}^{t,k-1} - w^* - \eta_{\text{local}} \cdot \overline{g}^{t,k-1} + 1_{\{k=0\}}\eta_{\text{local}} \cdot h^t\|_2^2$$

$$= \|\overline{u}^{t,k-1} - w^*\|_2^2 + \eta_{\text{local}}^2 \cdot \|\overline{g}^{t,k-1}\|_2^2 - 2\eta_{\text{local}}\langle \overline{u}^{t,k-1} - w^*, \overline{g}^{t,k-1}\rangle$$

$$+ 2\eta_{\text{local}} 1_{\{k=0\}}\langle \overline{u}^{t,k-1} - w^*, h^t\rangle - 2\eta_{\text{local}}^2 1_{\{k=0\}}\langle \overline{g}^{t,k-1}, h^t\rangle$$

$$+ \eta_{\text{local}}^2 1_{\{k=0\}} \cdot \|h^t\|_2^2 \tag{18}$$

Note by Theorem 4.2, we have:

$$\mathbb{E}[\text{desk}_t(\text{sk}_t(h))] = h, \qquad \mathbb{E}[\|\text{desk}_t(\text{sk}_t(h))\|_2^2] \leq (1 + \alpha) \cdot \|h\|_2^2$$

hold for any vector $h$. Hence, by taking expectation over Eq. (18),

$$\mathbb{E}[\|\overline{u}^{t,k} - w^*\|_2^2 | \mathcal{F}_t] = \mathbb{E}[\|\overline{u}^{t,k-1} - w^*\|_2^2 | \mathcal{F}_t] + \eta_{\text{local}}^2 \cdot \mathbb{E}[\|\overline{g}^{t,k-1}\|_2^2 | \mathcal{F}_t]$$

$$- 2\eta_{\text{local}}\,\mathbb{E}[\langle \overline{u}^{t,k-1} - w^*, \overline{g}^{t,k-1}\rangle | \mathcal{F}_t] + 1_{\{k=0\}} \cdot \eta_{\text{local}}^2 \cdot \mathbb{E}[\|h^t\|_2^2 | \mathcal{F}_t]$$

Note that since $\mathbb{E}[h^t | \mathcal{F}_t] = 0$, so the two inner products involving $h^t$ vanishes.

Since

$$\mathbb{E}[\|h^t\|_2^2 | \mathcal{F}_t] = \mathbb{E}[\|(I_d - \text{desk}_t \circ \text{sk}_t)(\sum_{i=0}^{K-1} \overline{g}^{t-1,i})\|_2^2 | \mathcal{F}_t]$$

$$\leq \alpha\,\mathbb{E}[\|\sum_{i=0}^{K-1} \overline{g}^{t-1,i}\|_2^2 | \mathcal{F}_t]$$

$$\leq \alpha K \sum_{i=0}^{K-1} \mathbb{E}[\|\overline{g}^{t-1,i}\|_2^2 | \mathcal{F}_t]$$

Taking total expectation, we have

$$\mathbb{E}[\|\overline{u}^{t,k} - w^*\|_2^2]$$

$$\leq \mathbb{E}[\|\overline{u}^{t,k-1} - w^*\|_2^2] + \eta_{\text{local}}^2 \cdot \mathbb{E}[\|\overline{g}^{t,k-1}\|_2^2] - 2\eta_{\text{local}} \mathbb{E}[\langle \overline{u}^{t,k-1} - w^*, \overline{g}^{t,k-1} \rangle]$$

$$+ 1_{\{k=0\}} \cdot \eta_{\text{local}}^2 \cdot \alpha K \sum_{i=0}^{K-1} \mathbb{E}[\|\overline{g}^{t-1,i}\|_2^2]$$

$$\leq \mathbb{E}[\|\overline{u}^{t,k-1} - w^*\|_2^2] + \eta_{\text{local}}^2 \cdot \mathbb{E}[2L^2 V^{t,k-1} + 4L(f(\overline{u}^{t,k-1}) - f(w^*))]$$

$$- 2\eta_{\text{local}} \mathbb{E}[f(\overline{u}^{t,k-1}) - f(w^*) - \frac{L}{2} V^{t,k-1} + \frac{\mu}{2} \|\overline{u}^{t,k-1} - w^*\|_2^2]$$

$$+ 1_{\{k=0\}} \cdot \eta_{\text{local}}^2 \cdot \alpha K \sum_{i=0}^{K-1} \mathbb{E}[2L^2 V^{t-1,i} + 4L(f(\overline{u}^{t-1,i}) - f(w^*))]$$

$$\leq (1 - \mu\eta_{\text{local}}) \mathbb{E}[\|\overline{u}^{t,k-1} - w^*\|_2^2] + \eta_{\text{local}} \cdot L \cdot (1 + 2\eta_{\text{local}}L) \cdot \mathbb{E}[V^{t,k-1}]$$

$$- 2\eta_{\text{local}} \cdot (1 - 2\eta_{\text{local}}L) \cdot \mathbb{E}[f(\overline{u}^{t,k-1}) - f(w^*)]$$

$$+ 1_{\{k=0\}} \cdot \eta_{\text{local}}^2 \cdot \alpha K \cdot \left( 2L^2 \sum_{i=0}^{K-1} \mathbb{E}[V^{t-1,i}] + 4L \sum_{i=0}^{K-1} \mathbb{E}[f(\overline{u}^{t-1,i}) - f(w^*)] \right)$$

where the second step follows from Lemma F.3 and Lemma F.4. Since $\eta_{\text{local}} \leq \frac{1}{4L}$, we have

$$\mathbb{E}[\|\overline{u}^{t,k} - w^*\|_2^2] \leq (1 - \mu\eta_{\text{local}}) \mathbb{E}[\|\overline{u}^{t,k-1} - w^*\|_2^2] + \frac{3}{2}\eta_{\text{local}} L \mathbb{E}[V^{t,k-1}] - \eta_{\text{local}} \mathbb{E}[f(\overline{u}^{t,k-1}) - f(w^*)]$$

$$+ 1_{\{k=0\}} \eta_{\text{local}}^2 \alpha K \left( 2L^2 \sum_{i=0}^{K-1} \mathbb{E}[V^{t-1,i}] + 4L \sum_{i=0}^{K-1} \mathbb{E}[f(\overline{u}^{t-1,i}) - f(w^*)] \right)$$

$$\square$$

## F.7 MAIN RESULT: CONVEX CASE

**Theorem F.7.** *Assume each $f_c$ is convex and $L$-smooth. If Theorem 4.2 holds and $\eta_{\text{local}} \leq \frac{1}{8(1+\alpha)LK}$,*

$$\mathbb{E}[f(\overline{w}^T) - f(w^*)] \leq \frac{4 \mathbb{E}[\|w^0 - w^*\|_2^2]}{\eta_{\text{local}} KT} + 32\eta_{\text{local}}^2 LK^2\sigma^2,$$

*where $\overline{w}^T = \frac{1}{KT}(\sum_{t=1}^{T} \sum_{k=0}^{K-1} \overline{u}^{t,k})$ is the average over parameters throughout the execution of Algorithm 1.*

*Proof.* Summing up Lemma F.6 as $t$ varies from 1 to $T$ and $k$ varies from 0 to $K-1$,

$$\mathbb{E}[\|\overline{u}^{T+1,0} - w^*\|_2^2] - \mathbb{E}[\|w^0 - w^*\|_2^2]$$

$$\leq \frac{3}{2}\eta_{\text{local}} L \sum_{t=1}^{T} \sum_{k=0}^{K-1} \mathbb{E}[V^{t,k}] - \eta_{\text{local}} \sum_{t=1}^{T} \sum_{k=0}^{K-1} \mathbb{E}[f(\overline{u}^{t,k}) - f(w^*)]$$

$$+ \sum_{t=1}^{T} \sum_{k=0}^{K-1} 1_{\{k=0\}} \eta_{\text{local}}^2 \alpha K \left( 2L^2 \sum_{i=0}^{K-1} \mathbb{E}[V^{t,i}] + 4L \sum_{i=0}^{K-1} \mathbb{E}[f(\overline{u}^{t,i}) - f(w^*)] \right)$$

$$= \frac{3}{2}\eta_{\text{local}} L \sum_{t=1}^{T} \sum_{k=0}^{K-1} \mathbb{E}[V^{t,k}] - \eta_{\text{local}} \sum_{t=1}^{T} \sum_{k=0}^{K-1} \mathbb{E}[f(\overline{u}^{t,k}) - f(w^*)]$$

$$+ \eta_{\text{local}}^2 \alpha K \left( 2L^2 \sum_{t=1}^{T} \sum_{i=0}^{K-1} \mathbb{E}[V^{t,i}] + 4L \sum_{t=1}^{T} \sum_{i=0}^{K-1} \mathbb{E}[f(\overline{u}^{t,i}) - f(w^*)] \right)$$

$$= \eta_{\text{local}} L (\frac{3}{2} + 2\eta_{\text{local}} \alpha LK) \sum_{t=1}^{T} \sum_{k=0}^{K-1} \mathbb{E}[V^{t,k}]$$

$$- \eta_{\text{local}}(1 - 4\eta_{\text{local}} \alpha LK) \sum_{t=1}^{T} \sum_{k=0}^{K-1} \mathbb{E}[f(\overline{u}^{t,k}) - f(w^*)]$$

$$\leq 2\eta_{\text{local}}L\sum_{t=1}^{T}\sum_{k=0}^{K-1}\mathbb{E}[V^{t,k}] - \frac{1}{2}\eta_{\text{local}}\sum_{t=1}^{T}\sum_{k=0}^{K-1}\mathbb{E}[f(\overline{u}^{t,k}) - f(w^*)]$$

$$\leq 2\eta_{\text{local}}L\sum_{t=1}^{T}(8\eta_{\text{local}}^2LK^2\sum_{i=0}^{K-1}\mathbb{E}[f(\overline{u}^{t,i}) - f(w^*)] + 4\eta_{\text{local}}^2K^3\sigma^2) - \frac{1}{2}\eta_{\text{local}}\sum_{t=1}^{T}\sum_{k=0}^{K-1}\mathbb{E}[f(\overline{u}^{t,k}) - f(w^*)]$$

$$\leq -\frac{1}{4}\eta_{\text{local}}\sum_{t=1}^{T}\sum_{k=0}^{K-1}\mathbb{E}[f(\overline{u}^{t,k}) - f(w^*)] + 8\eta_{\text{local}}^3LK^3T\sigma^2$$

where the fourth step follows from $\eta_{\text{local}} \leq \frac{1}{8\alpha LK}$, the last step follows from $\eta_{\text{local}} \leq \frac{1}{8LK}$. Rearranging the terms, we obtain

$$\frac{1}{KT}\sum_{t=1}^{T}\sum_{k=0}^{K-1}\mathbb{E}[f(\overline{u}^{t,k}) - f(w^*)] \leq \frac{4\,\mathbb{E}[\|w^0 - w^*\|_2^2]}{\eta_{\text{local}}KT} + 32\eta_{\text{local}}^2LK^2\sigma^2$$

Finally, by the convexity of $f$ we complete the proof. $\qquad\square$

Now we are ready to answer the question: how much communication cost is sufficient to guarantee $\mathbb{E}[f(\overline{w}^T) - f(w^*)] \leq \epsilon$? we have the following communication cost result:

**Theorem F.8.** *Assume each $f_c$ is convex and $L$-smooth. If Theorem 4.2 holds. With $O\left(\mathbb{E}[\|w^0 - w^*\|_2^2]N\max\{\frac{Ld}{\epsilon}, \frac{\sigma\sqrt{L}}{\epsilon^{3/2}}\}\right)$ bits of communication cost, Algorithm 1 outputs an $\epsilon$-optimal solution $\overline{w}^T$ satisfying:*

$$\mathbb{E}[f(\overline{w}^T) - f(w^*)] \leq \epsilon,$$

*where $\overline{w}^T = \frac{1}{KT}(\sum_{t=1}^{T}\sum_{k=0}^{K-1}\overline{u}^{t,k})$.*

*Proof.* To calculate the communication complexity, we first note communication only happens in sync steps. Specifically, in each sync step, the algorithm requires $O(Nb_{\text{sketch}})$ bits of communication cost, where $b_{\text{sketch}}$ denotes the sketching dimension. Therefore, the total cost of communication is given by $O(Nb_{\text{sketch}}T)$. To obtain the optimal communication cost for $\epsilon$-optimal solution, we choose $T, K, \eta_{\text{local}}$ and $b_{\text{sketch}}$ by solving the following optimization problem:

$$\min_{T,K,\eta_{\text{local}},b_{\text{sketch}},\alpha} Nb_{\text{sketch}}T$$

$$\text{s.t.} \quad 0 < \eta_{\text{local}} \leq \frac{1}{8(1+\alpha)LK}$$

$$\frac{4\,\mathbb{E}[\|w^0 - w^*\|_2^2]}{\eta_{\text{local}}KT} \leq \frac{\epsilon}{2}$$

$$32\eta_{\text{local}}^2LK^2\sigma^2 \leq \frac{\epsilon}{2}$$

$$d \geq b_{\text{sketch}} = O(\frac{d}{\alpha}) \geq 1$$

where $d$ is the parameter dimension and the last constraint is due to Theorem 4.2. Above constraints imply:

$$T \geq \frac{8\,\mathbb{E}[\|w^0 - w^*\|_2^2]}{\eta_{\text{local}}K\epsilon}, \quad , K\eta_{\text{local}} \leq \min\{\frac{1}{8(1+\alpha)L}, \frac{1}{8\sigma}\sqrt{\frac{\epsilon}{L}}\}$$

Therefore, when $\epsilon \geq \frac{\sigma^2}{(1+\alpha)^2L}$, the optimal solution is given by

$$K\eta_{\text{local}} = \frac{1}{8(1+\alpha)L}, \quad T = \frac{64\,\mathbb{E}[\|w^0 - w^*\|_2^2](1+\alpha)L}{\epsilon}, \quad b_{\text{sketch}} = O(\frac{d}{\alpha})$$

and the corresponding optimal communication cost is $O(\frac{\mathbb{E}[\|w^0 - w^*\|_2^2LNd}{\epsilon})$.

when $\epsilon < \frac{\sigma^2}{(1+\alpha)^2 L}$, the optimal solution is given by

$$K\eta_{\text{local}} = \frac{1}{8\sigma}\sqrt{\frac{\epsilon}{L}}, \ T = \frac{64\,\mathbb{E}[\|w^0 - w^*\|_2^2 \sigma\sqrt{L}]}{\epsilon^{3/2}}, \ b_{\text{sketch}} = O(\frac{d}{\alpha})$$

and the corresponding optimal communication cost is $O(\frac{\mathbb{E}[\|w^0 - w^*\|_2^2 \sigma\sqrt{L}Nd]}{\alpha\epsilon^{3/2}})$.

Combining above two cases, the optimal $\alpha$ is given by $O(d)$, and the corresponding optimal communication cost will be $O(\mathbb{E}[\|w^0 - w^*\|_2^2]N \max\{\frac{Ld}{\epsilon}, \frac{\sigma\sqrt{L}}{\epsilon^{3/2}}\})$. $\qquad\square$

## F.8 MAIN RESULT: STRONGLY-CONVEX CASE

**Theorem F.9.** *Assume each $f_c$ is $\mu$-strongly convex and $L$-smooth. If Theorem 4.2 holds and $\eta_{\text{local}} \leq \frac{1}{8(1+\alpha)LK}$,*

$$\mathbb{E}[f(w^{T+1}) - f(w^*)] \leq \frac{L}{2}\,\mathbb{E}[\|w^0 - w^*\|_2^2]e^{-\mu\eta_{\text{local}}T} + 4\eta_{\text{local}}^2 L^2 K^3 \sigma^2/\mu.$$

*Proof.* Summing up Lemma F.6 as $k$ varies from 0 to $K - 1$, then we have for any $t \geq 1$,

$$(\mathbb{E}[\|\overline{u}^{t+1,0} - w^*\|_2^2] + \sum_{k=1}^{K-1}\mathbb{E}[\|\overline{u}^{t,k} - w^*\|_2^2]) - (1 - \mu\eta_{\text{local}})\sum_{k=0}^{K-1}\mathbb{E}[\|\overline{u}^{t,k} - w^*\|_2^2])$$

$$\leq \frac{3}{2}\eta_{\text{local}}L\sum_{k=0}^{K-1}\mathbb{E}[V^{t,k}] - \eta_{\text{local}}\sum_{k=0}^{K-1}\mathbb{E}[f(\overline{u}^{t,k}) - f(w^*)]$$

$$+ \sum_{k=0}^{K-1}1_{\{k=0\}}\eta_{\text{local}}^2\alpha K\Big(2L^2\sum_{i=0}^{K-1}\mathbb{E}[V^{t,i}] + 4L\sum_{i=0}^{K-1}\mathbb{E}[f(\overline{u}^{t,i}) - f(w^*)]\Big)$$

$$= \frac{3}{2}\eta_{\text{local}}L\sum_{k=0}^{K-1}\mathbb{E}[V^{t,k}] - \eta_{\text{local}}\sum_{k=0}^{K-1}\mathbb{E}[f(\overline{u}^{t,k}) - f(w^*)]$$

$$+ \eta_{\text{local}}^2\alpha K\Big(2L^2\sum_{i=0}^{K-1}\mathbb{E}[V^{t,i}] + 4L\sum_{i=0}^{K-1}\mathbb{E}[f(\overline{u}^{t,i}) - f(w^*)]\Big)$$

$$= \eta_{\text{local}}L(\frac{3}{2} + 2\eta_{\text{local}}\alpha LK)\sum_{k=0}^{K-1}\mathbb{E}[V^{t,k}] - \eta_{\text{local}}(1 - 4\eta_{\text{local}}\alpha LK)\sum_{k=0}^{K-1}\mathbb{E}[f(\overline{u}^{t,k}) - f(w^*)]$$

$$\leq 2\eta_{\text{local}}L\sum_{k=0}^{K-1}\mathbb{E}[V^{t,k}] - \frac{1}{2}\eta_{\text{local}}\sum_{k=0}^{K-1}\mathbb{E}[f(\overline{u}^{t,k}) - f(w^*)]$$

$$\leq 2\eta_{\text{local}}L(8\eta_{\text{local}}^2 LK^2\sum_{i=0}^{K-1}\mathbb{E}[f(\overline{u}^{t,i}) - f(w^*)] + 4\eta_{\text{local}}^2 K^3 \sigma^2) - \frac{1}{2}\eta_{\text{local}}\sum_{k=0}^{K-1}\mathbb{E}[f(\overline{u}^{t,k}) - f(w^*)]$$

$$\leq -\frac{1}{4}\eta_{\text{local}}\sum_{k=0}^{K-1}\mathbb{E}[f(\overline{u}^{t,k}) - f(w^*)] + 8\eta_{\text{local}}^3 LK^3 \sigma^2$$

where the fourth step follows from $\eta_{\text{local}} \leq \frac{1}{8\alpha LK}$, the last step follows from $\eta_{\text{local}} \leq \frac{1}{8LK}$. Rearranging the terms, we obtain

$$\mathbb{E}[\|\overline{u}^{t+1,0} - w^*\|_2^2] \leq (1 - \mu\eta_{\text{local}})\mathbb{E}[\|\overline{u}^{t,0} - w^*\|_2^2] + 8\eta_{\text{local}}^3 LK^3 \sigma^2$$

implying

$$\mathbb{E}[\|\overline{u}^{t+1,0} - w^*\|_2^2] - 8\eta_{\text{local}}^2 LK^3 \sigma^2/\mu \leq (1 - \mu\eta_{\text{local}})(\mathbb{E}[\|\overline{u}^{t,0} - w^*\|_2^2] - 8\eta_{\text{local}}^2 LK^3 \sigma^2/\mu).$$

Therefore, we have

$$\mathbb{E}[\|w^{T+1} - w^*\|_2^2] - 8\eta_{\text{local}}^2 LK^3 \sigma^2/\mu \leq (1 - \mu\eta_{\text{local}})^T(\mathbb{E}[\|w^0 - w^*\|_2^2] - 8\eta_{\text{local}}^2 LK^3 \sigma^2/\mu)$$

$$\leq \mathbb{E}[\|w^0 - w^*\|_2^2]e^{-\mu\eta_{\text{local}}T}$$

Finally, by $L$-smoothness of function $f$, we obtain

$$\mathbb{E}[f(w^{T+1}) - f(w^*)] \leq \frac{L}{2}\mathbb{E}[\|w^{T+1} - w^*\|_2^2] \leq \frac{L}{2}\mathbb{E}[\|w^0 - w^*\|_2^2]e^{-\mu\eta_{\text{local}}T} + 4\eta_{\text{local}}^2 L^2 K^3 \sigma^2/\mu.$$

$\square$

**Theorem F.10.** *Assume each $f_c$ is $\mu$-strongly convex and $L$-smooth. If Theorem 4.2 holds. With $O\left(\frac{LN}{\mu}\max\{d, \sqrt{\frac{\sigma^2}{\mu\epsilon}}\}\log(\frac{L\,\mathbb{E}[\|w^0-w^*\|_2^2]}{\epsilon})\right)$ bits of communication cost, Algorithm 1 outputs an $\epsilon$-optimal solution $w^T$ satisfying:*

$$\mathbb{E}[f(w^T) - f(w^*)] \leq \epsilon.$$

*Proof.* To calculate the communication complexity, we first note communication only happens in sync steps. Specifically, in each sync step, the algorithm requires $O(Nb_{\text{sketch}})$ bits of communication cost, where $b_{\text{sketch}}$ denotes the sketching dimension. Therefore, the total cost of communication is given by $O(Nb_{\text{sketch}}T)$. To obtain the optimal communication cost for $\epsilon$-optimal solution, we choose $T, K, \eta_{\text{local}}$ and $b_{\text{sketch}}$ by solving the following optimization problem:

$$\min_{T,K,\eta_{\text{local}},b_{\text{sketch}},\alpha} Nb_{\text{sketch}}T$$

$$\text{s.t.} \quad 0 < \eta_{\text{local}} \leq \frac{1}{8(1+\alpha)LK}$$

$$\frac{L}{2}\mathbb{E}[\|w^0 - w^*\|_2^2]e^{-\mu\eta_{\text{local}}T} \leq \frac{\epsilon}{2}$$

$$4\eta_{\text{local}}^2 L^2 K^3 \sigma^2/\mu \leq \frac{\epsilon}{2}$$

$$d \geq b_{\text{sketch}} = O(\frac{d}{\alpha}) \geq 1$$

where $d$ is the parameter dimension and the last constraint is due to Theorem 4.2. Above constraints imply:

$$T \geq \frac{1}{\mu\eta_{\text{local}}}\log(\frac{L\,\mathbb{E}[\|w^0 - w^*\|_2^2]}{\epsilon}), \quad \eta_{\text{local}} \leq \min\{\frac{1}{8(1+\alpha)LK}, \frac{1}{2LK\sigma}\sqrt{\frac{\mu\epsilon}{2K}}\}$$

Therefore, the optimal value is given when $K = 1$. When $\epsilon \geq \frac{\sigma^2}{16(1+\alpha)^2\mu}$, the optimal solution is given by

$$\eta_{\text{local}} = \frac{1}{8(1+\alpha)L}, \; T = \frac{8(1+\alpha)L}{\mu}\log(\frac{L\,\mathbb{E}[\|w^0 - w^*\|_2^2]}{\epsilon}), \; b_{\text{sketch}} = O(\frac{d}{\alpha})$$

and the corresponding optimal communication cost is $O(\frac{LNd}{\mu}\log(\frac{L\,\mathbb{E}[\|w^0-w^*\|_2^2]}{\epsilon}))$.

when $\epsilon < \frac{\sigma^2}{16(1+\alpha)^2\mu}$, the optimal solution is given by

$$\eta_{\text{local}} = \frac{1}{2L\sigma}\sqrt{\frac{\mu\epsilon}{2}}, \; T = \frac{2L\sigma}{\mu^{3/2}}\sqrt{\frac{2}{\epsilon}}\log(\frac{L\,\mathbb{E}[\|w^0 - w^*\|_2^2]}{\epsilon}), \; b_{\text{sketch}} = O(\frac{d}{\alpha})$$

and the corresponding optimal communication cost is $O(\frac{\sigma LNd}{\alpha\mu^{3/2}\sqrt{\epsilon}}\log(\frac{L\,\mathbb{E}[\|w^0-w^*\|_2^2]}{\epsilon}))$.

Combining above two cases, the optimal $\alpha$ is given by $O(d)$, and the corresponding optimal communication cost will be $O(\frac{LN}{\mu}\max\{d, \sqrt{\frac{\sigma^2}{\mu\epsilon}}\}\log(\frac{L\,\mathbb{E}[\|w^0-w^*\|_2^2]}{\epsilon}))$. $\square$

# G $k$-STEP NON-CONVEX $f$ CONVERGENCE ANALYSIS

In this section, we present convergence result for non-convex $f$ case in the $k$-local-step regime. In order for the proof to go through, we assume that for any $c \in [N]$ and any $w \in \mathbb{R}^d$, there exists a universal constant $G$ such that

$$\|\nabla f_c(w)\|_2 \leq G.$$

Throughout the proof, we will use $\mathcal{F}_t$ to denote the sequence $w_{t-1}, w_{t-2}, \ldots, w_0$. Also, we use $\eta$ as a shorthand for $\eta_{\text{global}} \cdot \eta_{\text{local}}$.

Note that in $k$-local-step scheme, the average of local gradients is no longer the true gradient, therefore, we can no longer bound everything using the true gradients. This means it's necessary to introduce the gradient norm upper bound assumption.

**Lemma G.1.** *Let* $f : \mathbb{R}^d \to \mathbb{R}$ *satisfying Assumption 3.1 and* sk/desk *functions satisfying Theorem 4.2. Further, assume* $\eta_{\text{local}} \leq \frac{1}{2LK}$. *Then*

$$\mathbb{E}[f(w^{t+1}) - f(w^t) \mid \mathcal{F}_t] \leq -\eta_{\text{global}} \cdot \|\nabla f(w^t)\|_2^2 + \eta \cdot L \cdot K^2 \cdot G^2 \cdot \left(\eta_{\text{local}} + \frac{\eta}{2} \cdot (1 + \alpha)\right)$$

*Proof.* We start by bounding $f(w^{t+1}) - f(w^t)$ without taking conditional expectation:

$$f(w^{t+1}) - f(w^t)$$
$$\leq \langle w^{t+1} - w^t, \nabla f(w^t)\rangle + \frac{L}{2}\|w^{t+1} - w^t\|_2^2$$
$$= \langle \text{desk}_t(\Delta \widetilde{w}^t), \nabla f(w^t)\rangle + \frac{L}{2}\|\text{desk}_t(\Delta \widetilde{w}^t)\|_2^2$$
$$= A + \frac{L}{2}B$$

where

$$A := -\langle \eta_{\text{global}} \cdot \text{desk}_t(\frac{1}{N}\sum_{c=1}^{N}\text{sk}_t(\sum_{k=0}^{K-1}\eta_{\text{local}} \cdot \nabla f_c(u_c^{t,k}))), \nabla f(w^t)\rangle$$

$$B := \|\eta_{\text{global}} \cdot \text{desk}_t(\frac{1}{N}\sum_{c=1}^{N}\text{sk}_t(\sum_{k=1}^{K}\eta_{\text{local}} \cdot \nabla f_c(u_c^{t,k})))\|_2^2$$

**Bounding** $\mathbb{E}[A \mid \mathcal{F}_t]$   Using the fact that $\text{sk}_t/\text{desk}_t$ are linear functions and $\mathbb{E}[\text{desk}_t(\text{sk}_t(h))] = h$, we get

$$\mathbb{E}[A \mid \mathcal{F}_t] = -\langle \eta_{\text{global}} \cdot \frac{1}{N}\sum_{c=1}^{N}\sum_{k=0}^{K-1}\eta_{\text{local}} \cdot \nabla f_c(u_c^{t,k}), \nabla f(w^t)\rangle$$

$$= -\eta_{\text{global}} \cdot \langle \frac{1}{N}\sum_{c=1}^{N}\Big(\sum_{k=0}^{K-1}\eta_{\text{local}} \cdot \nabla f_c(u_c^{t,k}) - \nabla f_c(w^t) + \nabla f_c(w^t)\Big), \nabla f(w^t)\rangle$$

$$= -\eta_{\text{global}} \cdot \|\nabla f(w^t)\|_2^2 + \eta_{\text{global}} \cdot \eta_{\text{local}} \cdot \frac{1}{N}\sum_{c=1}^{N}\sum_{k=0}^{K-1}\langle \nabla f_c(u_c^{t,k}) - \nabla f_c(w^t), \nabla f(w^t)\rangle$$

It suffices to bound the inner product, notice for $k = 0$, the inner product is 0, so assume $k \geq 1$:

$$\langle \nabla f_c(u_c^{t,k}) - \nabla f_c(w^t), \nabla f(w^t)\rangle$$
$$\leq \|\nabla f_c(u_c^{t,k}) - \nabla f_c(w^t)\|_2 \cdot \|\nabla f(w^t)\|_2$$
$$\leq L \cdot \|u_c^{t,k} - w^t\|_2 \cdot \|\nabla f(w^t)\|_2 \tag{19}$$

where the gap between $u_c^{t,k}$ and $w^t$ can be further expanded:

$$\|u_c^{t,k} - w^t\|_2 = \|u_c^{t,k} - u_0^{t,k}\|_2$$
$$= \|\eta_{\text{local}}\sum_{i=0}^{k-1}\nabla f_c(u_c^{t,i})\|_2$$
$$\leq \eta_{\text{local}}\sum_{i=0}^{k-1}\|\nabla f_c(u_c^{t,i})\|_2$$

$$\leq \eta_{\text{local}} \cdot k \cdot G \tag{20}$$

Plug in Eq. (20) to Eq. (19), we get

$$\langle \nabla f_c(u_c^{t,k}) - \nabla f_c(w^t), \nabla f(w^t) \rangle \leq L \cdot \eta_{\text{local}} \cdot k \cdot G^2$$

Recall that $\eta = \eta_{\text{global}} \cdot \eta_{\text{local}}$. Put things together, we finally obtain a bound on $\mathbb{E}[A \mid \mathcal{F}_t]$:

$$\mathbb{E}[A \mid \mathcal{F}_t] \leq -\eta_{\text{global}} \cdot \|\nabla f(w^t)\|_2^2 + \eta \cdot \eta_{\text{local}} \cdot L \cdot \left( \sum_{k=0}^{K-1} k \right) \cdot G^2$$

$$\leq -\eta_{\text{global}} \cdot \|\nabla f(w^t)\|_2^2 + \eta \cdot \eta_{\text{local}} \cdot L \cdot K^2 \cdot G^2 \tag{21}$$

**Bounding $\mathbb{E}[B \mid \mathcal{F}_t]$**  Using the fact that $\mathsf{sk}_t/\mathsf{desk}_t$ are linear functions, we get

$$B = \eta_{\text{global}}^2 \cdot \eta_{\text{local}}^2 \cdot \frac{1}{N^2} \cdot \| \sum_{c=1}^{N} \sum_{k=0}^{K-1} \mathsf{desk}_t(\mathsf{sk}_t(\nabla f_c(u_c^{t,k}))) \|_2^2$$

$$\leq \eta_{\text{global}}^2 \cdot \eta_{\text{local}}^2 \cdot \frac{1}{N^2} \cdot N \cdot K \sum_{c=1}^{N} \sum_{k=0}^{K-1} \cdot \|\mathsf{desk}_t(\mathsf{sk}_t(\nabla f_c(u_c^{t,k}))) \|_2^2$$

$$= \eta^2 \cdot \frac{K}{N} \cdot \sum_{c=1}^{N} \sum_{k=0}^{K-1} \|\mathsf{desk}_t(\mathsf{sk}_t(\nabla f_c(u_c^{t,k}))) \|_2^2$$

Using variance bound of $\mathsf{desk}_t(\mathsf{sk}_t(h))$, we get

$$\mathbb{E}[B \mid \mathcal{F}_t] \leq \eta^2 \cdot \frac{K}{N} \cdot (1 + \alpha) \cdot \sum_{c=1}^{N} \sum_{k=0}^{K-1} \|\nabla f_c(u_c^{t,k})\|_2^2$$

$$\leq \eta^2 \cdot \frac{K}{N} \cdot (1 + \alpha) \cdot \sum_{c=1}^{N} \sum_{k=0}^{K-1} G^2$$

$$= \eta^2 \cdot K^2 \cdot (1 + \alpha) \cdot G^2 \tag{22}$$

**Put things together**  Put the bound on $\mathbb{E}[A \mid \mathcal{F}_t]$ and the bound on $\mathbb{E}[B \mid \mathcal{F}_t]$, we get

$$\mathbb{E}[f(w^{t+1}) - f(w^t) \mid \mathcal{F}_t]$$

$$\leq -\eta_{\text{global}} \cdot \|\nabla f(w^t)\|_2^2 + \eta \cdot \eta_{\text{local}} \cdot L \cdot K^2 \cdot G^2 + \frac{L}{2} \cdot \eta^2 \cdot K^2 \cdot (1 + \alpha) \cdot G^2$$

$$= -\eta_{\text{global}} \cdot \|\nabla f(w^t)\|_2^2 + \eta \cdot L \cdot K^2 \cdot G^2 \cdot \left( \eta_{\text{local}} + \frac{\eta}{2} \cdot (1 + \alpha) \right) \qquad \square$$

**Theorem G.2.** *Let $f : \mathbb{R}^d \to \mathbb{R}$ be $L$-smooth. Let $w^* \in \mathbb{R}^d$ be the optimal solution to $f$ and assume* $\mathsf{sk}/\mathsf{desk}$ *functions satisfying Theorem 4.2. Then*

$$\min_{t \in [T]} \mathbb{E}[\|\nabla f(w^t)\|_2^2] \leq \frac{1}{(T+1)\eta_{\text{global}}} \cdot (\mathbb{E}[f(w^0)] - f(w^*)) + \eta_{\text{local}} \cdot LK^2G^2 \cdot \left( \eta_{\text{local}} + \frac{\eta}{2} \cdot (1 + \alpha) \right)$$

*Proof.* By Lemma G.1, we know that

$$\mathbb{E}[f(w^{t+1}) \mid \mathcal{F}_t] - f(w^t) \leq -\eta_{\text{global}} \cdot \|\nabla f(w^t)\|_2^2 + \eta \cdot L \cdot K^2 \cdot G^2 \cdot \left( \eta_{\text{local}} + \frac{\eta}{2} \cdot (1 + \alpha) \right)$$

Rearranging the inequality and taking total expectation, we get

$$\mathbb{E}[\|\nabla f(w^t)\|_2^2] \leq \frac{1}{\eta_{\text{global}}} \cdot (\mathbb{E}[f(w^t)] - \mathbb{E}[f(w^{t+1})]) + \eta_{\text{local}} \cdot LK^2G^2 \cdot \left( \eta_{\text{local}} + \frac{\eta}{2} \cdot (1 + \alpha) \right)$$

Sum over all $T$ iterations and averaging, we arrive at

$$\frac{1}{T+1} \sum_{t=0}^{T} \mathbb{E}[\|\nabla f(w^t)\|_2^2]$$

$$\leq \frac{1}{(T+1)\eta_{\text{global}}} \cdot (\mathbb{E}[f(w^0)] - \mathbb{E}[f(w^T)]) + \eta_{\text{local}} \cdot LK^2G^2 \cdot \left(\eta_{\text{local}} + \frac{\eta}{2} \cdot (1+\alpha)\right)$$

$$\leq \frac{1}{(T+1)\eta_{\text{global}}} \cdot (\mathbb{E}[f(w^0)] - f(w^*)) + \eta_{\text{local}} \cdot LK^2G^2 \cdot \left(\eta_{\text{local}} + \frac{\eta}{2} \cdot (1+\alpha)\right)$$

$\square$

## H   DIFFERENTIAL PRIVACY

We define $(\epsilon, \delta)$-differential privacy Dwork et al. (2006b;a) as

**Definition H.1.** *Let $\epsilon, \delta$ be positive real number and $\mathcal{A}$ be a randomized algorithm that takes a dataset as input (representing the actions of the trusted party holding the data). Let $\text{im}(\mathcal{A})$ denote the image of $\mathcal{A}$. The algorithm $\mathcal{A}$ is said to provide $\epsilon, \delta$-differential privacy if, for all datasets $D_1$ and $D_2$ that differ on a single element (i.e., the data of one person), and all subsets $S$ of $\text{im}(\mathcal{A})$:*

$$\Pr[\mathcal{A}(D_1) \in S] \leq \exp(\epsilon) \cdot \Pr[\mathcal{A}(D_2) \in S] + \delta$$

*where the probability is taken over the randomness used by the algorithm.*

We state a lemma from prior work Kenthapadi et al. (2013),

**Lemma H.2** (Kenthapadi et al. (2013))**.** *Let $\delta \in (0, 1/4)$ and $\epsilon > 0$. Let $Y$ and $Y'$ be points in $\mathbb{R}^d$ such that $\|Y - Y'\|_2 \leq w$. Then for any $D \subset \mathbb{R}^d$, and any $\Delta$ drawn from $\{\mathcal{N}(0, \sigma^2)\}^d$, where $\sigma \geq 4w\epsilon^{-1}\sqrt{\log(1/\delta)}$, the following inequality holds:*

$$\Pr[Y' + \Delta \in D] \leq e^\epsilon \cdot \Pr[Y + \Delta \in D] + \delta.$$

Throughout this section, we assume that for any gradient $g_c$ and any one-entry perturbation $g_c'$ of $g_c$, the magnitude of the one-entry perturbation is upper bounded by $\gamma$.

Let $R \in \mathbb{R}^{b_{\text{sketch}} \times d}$ denote our sketching matrix.

Let $Y', Y \in \mathbb{R}^{b_{\text{sketch}}}$. Let $g, g' \in \mathbb{R}^d$ such that $g$ and $g'$ differ by exactly one entry and the magnitude of this difference is bounded by $\gamma$. Let $Y = Rg$ and $Y' = Rg'$.

We demonstrate how to pick the corresponding variance of noise for AMS sketch. We remark that for other sketching matrices where entries are bounded and hence its spectral norm is bounded, one can run similar argument to obtain such results.

**Fact H.3.** *If $R$ is an AMS sketching matrices, then*

$$\sqrt{d}/b \leq \|R\| \leq \sqrt{d}/\sqrt{b}.$$

*Proof.* Recall that each entry of an AMS sketching matrix is $\pm\frac{1}{\sqrt{b}}$, hence its max row or column $\ell_1$ norm is $\frac{d}{\sqrt{b}}$. Recall that max row or column $\ell_1$ norm is an upper bound on the spectral norm, hence we know that $\|R\| \leq \frac{d}{\sqrt{b}}$. On the other hand, we know its spectral norm is lower bounded by its max row or column $\ell_2$ norm, hence,

$$\|R\| \geq \|R\|_F/b \geq \sqrt{d}/b.$$

Put things together, we have

$$\sqrt{d}/b \leq \|R\| \leq \sqrt{d}/\sqrt{b}.$$

$\square$

Putting it all together, we have

**Lemma H.4.** *Let $\Delta \in \mathbb{R}^{b_{\text{sketch}}}$ denote the noise vector that sampled from Gaussian distribution $\mathcal{N}(0, \sigma^2 I_b)$ with $\sigma \geq 4\sqrt{\frac{d}{b_{\text{sketch}}}}\gamma\epsilon^{-1}\sqrt{\log(1/\delta)}$, then we have $Rg + \Delta$ is $(\epsilon, \delta)$-differential private.*

*Proof.* By Lemma H.2, it suffices to to bound $\|Y' - Y\|_2$. Note that

$$
\begin{aligned}
\|Y - Y'\|_2 &= \|Rg - Rg'\|_2 \\
&\leq \|R\| \cdot \|g - g'\|_2 \\
&\leq \sqrt{\frac{d}{b_{\text{sketch}}}} \|g - g'\|_2 \\
&\leq \sqrt{\frac{d}{b_{\text{sketch}}}} \gamma.
\end{aligned}
$$

This means if we pick $\sigma \geq 4\sqrt{\frac{d}{b_{\text{sketch}}}} \gamma \epsilon^{-1} \sqrt{\log(1/\delta)}$, then our sketched gradient with noise is $(\epsilon, \delta)$-differential private. $\square$

