# OpenReview forum: "Iterative Sketching and its Application to Federated Learning"
_ICLR.cc/2022/Conference — ICLR 2022 Submitted_

### Official Review · Reviewer_YCQM · 2021-10-28

**Correctness:** 3
**Technical Novelty And Significance:** 1
**Empirical Novelty And Significance:** Not applicable
**Recommendation:** 3
**Confidence:** 4

**Main Review:**

**A. Novelty in the update aggregation mechanism:**

A1. The idea of using random matrices to reduce communication complexity in federated learning is not new at all. See, for instance, the following papers (which the authors seem to be oblivious to):

   i) Suresh, A. T., Felix, X. Y., Kumar, S., & McMahan, H. B. (2017, July). Distributed mean estimation with limited communication. In International Conference on Machine Learning (pp. 3329-3337).

   ii) Mayekar, P., & Tyagi, H. (2020, June). RATQ: A universal fixed-length quantizer for stochastic optimization. In International Conference on Artificial Intelligence and Statistics (pp. 1399-1409).

   iii) Chen, W. N., Kairouz, P., & Özgür, A. (2020). Breaking the communication-privacy-accuracy trilemma. In advances in Neural Information Processing Systems (pp. 3312--3324).

  iv) Agarwal, N., Suresh, A. T., Yu, F., Kumar, S., & Mcmahan, H. B. (2018). cpSGD: Communication-efficient and differentially-private distributed SGD. arXiv preprint arXiv:1805.10559.

The lack of comparison of the author's proposed scheme with any of these papers is astonishing and is definitely warranted for a fair understanding of the authors' contribution.

A2. From my understanding of the papers mentioned above, there is nothing that the authors propose in their aggregating algorithm(Section 4) that wasn't already known.

     a. For instance, both i, ii, highlight, and crucially use the fact that when the updates are processed using certain random matrices, the coordinates of the processed vector have small absolute values; a property amenable for efficient vector quantization. The authors of this paper, too, point out a similar property in Definition 4.1.

      b. Similarly using $R^T$ to "de-sketch" the sketched update has also been highlighted in these papers.


A3.  I don't see the point behind definition 4.1, since the authors don't seem to exploit this property to build better communication protocols. In particular, since convergence guarantees seem to rely on only the second moment bound in T 4.2, something as unsophisticated as uniformly sampling $b_{sketch}$ coordinates would (and scaling appropriately and setting other coordinates to $0$) would give the same guarantees as sophisticated sketches (precisely the same as in T 4.2). What is the point of these sophisticated sketches then? This does not come out in the paper at all.

As I pointed out in A2., properties similar to Definition 4.1 have been highlighted by many papers since 2017. The rationale behind highlighting such properties in these papers was to further exploit such properties for efficient vector quantization.  It is pointless to use these sketches, without exploiting them to build efficient vector quantizers.


 **B. Presentation and clarity**

B.1 The authors can significantly improve the presentation and the clarity of this paper. The paper has many ambiguities and typos in its current form, and, frankly, it is far from being ready for publication.

B.2 In  a 43 page research paper whose main goal is to reduce the communication complexity of federated learning, it is astonishing that not a single line is used to describe the overall quantization procedure.

 Perhaps the author meant to use the default 32 bit (64 bit) precision to transmit each scalar. Even this is important to state explicitly. Of course, if this is indeed the case, then what is the point of using sophisticated sketches (See point A.3 for more details).

B.3 The parameter $\alpha$ is never defined. This parameter crucially shows up in convergence results. ( After going through the proofs, I could understand this was the variance blowup parameter. Nonetheless, it should have been highlighted in the first place.)

B.4 It is only in the discussion section that the authors point out that their convergence results build upon the work of Khaled 19. In my opinion, this should be explicitly stated in Section 5 and highlighted, where the convergence results are presented.

B.5 There are several inconsequential typos and grammatical mistakes throughout the paper. Thorough proofreading of the paper should be able to fix this.

B.6 It is disappointing that there are no labels to identify the parts in the appendix as proofs of the results presented in the main paper. Given the lack of such labels, the appendix section is extremely difficult to navigate through.

**Summary Of The Paper:**

The paper studies the application of linear sketching algorithms to reduce the communication complexity in federated Learning problems. In particular, the paper proposes that the clients send a low-dimensional linear sketch of their updates to the server. Convergence rate of local SGD when deployed with such a method for aggregating the updates are discussed for various function classes. Finally, the paper also discusses a method for integrating differentially privacy guarantees in their method.

**Summary Of The Review:**

GIven the lack of novelty in the paper's main algorithm (A.2), the lack of comparison with other published works using similar approaches (A.1, B.4), the fact that authors themselves show a lack of understanding of the motivation behind using sketches (A.3), and multiple presentation issues (B.1 - B.6), I  recommend the paper be rejected.

---

> ### Author Response · Authors · 2021-11-22
> **Thank you for your review and comments!**
>
> We thank the reviewer for the valuable comments and questions. We respond to them as follows:
>
> > It is astonishing that not a single line is used to describe the overall quantization procedure.
>
> We view quantization as an orthogonal direction of our sketching approach. Our approach tries to optimize the upper-level update procedure of the local (stochastic) gradient descent algorithm and is not conflicted with the quantization techniques. Similar to the work [1] mentioned by Reviewer RjL4, our methods can directly combine our algorithmic approach with advanced gradient quantization techniques, asynchronous updates, etc.
>
> > Uniform sampling will work and no sophisticated sampling is needed.
>
> Simple uniform sampling will not work here. As shown in Lemma D.16 in supplementary material, uniform sampling matrix has coordinate-wise embedding with parameter $a = O(d)$, which linearly scales with the dimension of the dataset, indicating the excess perturbation introduced due to using uniform sampling is too large to fit in our needs. While commonly used sketching matrices, such as random Gaussian, AMS, SRHT, has a $O(1)$ coordinate-wise embedding property, indicating they will introduce much smaller noise to the original updates.
>
> > Novelty of the paper.
>
> We note that while there are many works that use random matrices to reduce the dimension of gradient vectors and speedup the communication, our approaches are one of the most general, since it is compatible with almost all the popular sketching matrices, such as Count Sketch [2, 3], SRHT [4] and sparse embedding [5, 6, 7, 8]. Moreover, all of our sketch and de-sketch operations are linear, which is in contrast to using the Count Sketch matrix to sketch and top-k to de-sketch. We remark that our work resembles a similarity of using sketching for convex optimization tasks, such as linear programming [9] and empirical risk minimization [11], maximum weighted matching [12, 13], max-flow [12], linear programs with small tree-width [10, 14].
>
> [1] Communication-efficient Distributed SGD with Sketching, ICML 2020.
> [2] Charikar, Chen and Farach-Colten. Finding Frequent Items in Data Streams. ICALP 2002.
> [3] Clarkson and Woodruff. Low Rank Approximation and Regression in Input Sparsity Time. STOC 2013.
> [4] Ailon and Chazelle. Approximate Nearest Neighbors and the Fast Johnson-Lindenstrauss Transform. STOC 2006.
> [5] Dasgupta, Kumar and Sarlos. A Sparse Johnson--Lindenstrauss Transform. STOC 2010.
> [6] Kane and Nelson. Sparser Johnson-Lindenstrauss Transforms. JACM 2014.
> [7] Nelson and Nguyen. OSNAP: Faster numerical linear algebra algorithms via sparser subspace embeddings. FOCS 2013.
> [8] Cohen, Jayram and Nelson. Simple Analysis of the Sparse Johnson-Lindenstrauss Transform. SOSA 2018.
> [9] Jiang, Song, Weinstein and Zhang. Faster Dynamic Matrix Inverse for Faster LPs. STOC 2021.
> [10] Dong, Lee and Ye. A Nearly-Linear Time Algorithm for Linear Programs with Small Treewidth: A Multiscale Representation of Robust Central Path. STOC 2021.
> [11] Lee, Song and Zhang. Solving Empirical Risk Minimization in the Current Matrix Multiplication Time. COLT 2019.
> [12] Brand, Lee, Liu, Saranuarak, Sidford, Song and Wang. Minimum cost flows, MDPs, and L1-regression in nearly linear time for dense instances. STOC 2021.
> [13] Liu, Song and Zhang. Breaking the n-Pass Barrier: A Streaming Algorithm for Maximum Weight Bipartite Matching. 2021.
> [14] Ye. Fast Algorithm for Solving Structured Convex Programs. 2020. Undergraduate thesis at The University of Washington.

---

> > ### Comment · Reviewer_YCQM · 2021-11-29
> > **Response to the rebuttal**
> >
> > After reading the authors' rebuttal and the other reviews, I am happy with my original review.
> >
> > To further clarify my review, I will respond to one specific point in the authors' rebuttal
> > "Simple uniform sampling will not work here. As shown in Lemma D.16 in supplementary material, uniform sampling matrix has coordinate-wise embedding with parameter, which linearly scales with the dimension of the dataset, indicating the excess perturbation introduced due to using uniform sampling is too large to fit in our needs. While commonly used sketching matrices, such as random Gaussian, AMS, SRHT, has a coordinate-wise embedding property, indicating they will introduce much smaller noise to the original updates."
> >
> > This is precisely my point. Despite the fact that Uniform Sampling has a larger embedding parameter, it could attain the same theoretical convergence guarantees as in Theorem 4.2 (One where authors propose to use sophisticated sketches.)

---

> > > ### Author Response · Authors · 2021-11-29
> > > **Effect of embedding parameter to convergence analysis**
> > >
> > > Thanks for your response. We want to point out that however, by using uniform sampling, one has to pick a much smaller step size since step size is chosen as $\propto \frac{1}{\alpha}$ where $\alpha$ is the embedding parameter, in all other sketching matrices other than uniform sampling, $\alpha$ is a small constant (around 2 or 3), but for uniform sampling, $\alpha=d$, which means one would shrink the step size by a factor of $O(d)$. This has a significant impact on the convergence rate, as we have discussed towards the end of page 8 in our paper.

---

### Official Review · Reviewer_NaBa · 2021-10-30

**Correctness:** 4
**Technical Novelty And Significance:** 2
**Empirical Novelty And Significance:** 2
**Recommendation:** 5
**Confidence:** 3

**Main Review:**

Let me first summarize the main idea of this paper. The idea is conceptually simple, it uses a JL sketch to reduce the dimension of gradient, and since this *projected* gradient has large correlations with the original gradient, one can prove the convergence.


Strength. I found it interesting to combine sketching with federate learning. The idea is conceptually simple but no one explores it before.


Weakness. There is no experiments, and the total communication cost does not decrease.

**Summary Of The Paper:**

The paper proposes a sketching approach to reduce the communication cost of federate learning, and prove the convergence of their proposed method for convex function. By adding gaussian noise to the gradient, it can also guarantee different privacy.

**Summary Of The Review:**

---------------------------------
Post rebuttal:

I have read author's response as well as other reviewer's suggestion. I still think the paper is of value at certain aspects. But as other reviewers pointed out, I suggest the authors include a more detailed discussion on related work.

---

> ### Author Response · Authors · 2021-11-22
> **Thank you for your review and comments!**
>
> We thank the reviewer for the valuable comments and suggestions. We have conducted experiments on synthetic datasets and show the total communication cost is of the same order as in our theorems. We will continue to conduct more experiments on more complicated tasks and datasets.

---

### Official Review · Reviewer_RjL4 · 2021-11-01

**Correctness:** 2
**Technical Novelty And Significance:** 2
**Empirical Novelty And Significance:** Not applicable
**Recommendation:** 3
**Confidence:** 3

**Main Review:**

Federated learning has recently become a popular framework for training machine learning models. Reducing the communication cost of federated learning is an important problem. However the contribution of this paper is incremental. First, combining sketching with federated learning has been studied by several existing papers (see comment 5). Also, the paper doesn't provide any empirical results and the theoretical part is also on the weak side. Here are the details:


1. Some notations of different sections are inconsistent. For instance, the paper uses $a$ to denote the value of coordinate-wise embedding in section 4.1, but uses $\alpha$ instead in section 5 and appendix. I suggest the authors check the notations in the paper to increase the readability of the paper.

2. The theorems in section 5 only require Assumption 3.1 holds, but I think they also require theorem 4.2 holds since their proofs use the property of the sketching matrices. In addition, I'm confused with the proof of Lemma F.6. The inequality below Eq. (18) says $E[dest_t(sk_t(h))]\leq (1+\alpha)||h||^2$. However, Thm 4.2 only tells us $E[dest_t(sk_t(h))]\leq (1+\alpha d / b_{sketch})||h||^2$. The paper should provide some explanation on why it can remove the  $d / b_{sketch}$ term?

3. I'm also confused with the proof of Corollary 5.7 and 5.9 (Thm F.8 and Thm F.10). In the last sentence of the proof of Thm F.8 says "the optimal $\alpha$ is given by O(d)." Since $\alpha$ is the value of the coordinate-wise embedding of the sketching matrix, how to let $\alpha$ be O(d)? The proof of Thm F.10 faces the same problem.

4. For Thm 5.6 and 5.8, $\eta=1/(8(1+\alpha)LK)$ is not enough to guarantee $w^T$ converge to the $w^*$ since the last term in the bounds won't decrease when the number of iteration increase. I think $\eta$ should be related with $T$ or $\epsilon$.

5. It seems that the clients use different sketching matrices for different $t$. How to guarantee that all clients use the same sketching matrix in each round without communication?

6. The following papers also adopt sketching to reduce the communication cost of federated learning. I suggest the author discuss the difference between those papers and this work.

"Communication-efficient Distributed SGD with Sketching", NeurIPS 2019.

"FetchSGD: Communication-Efficient Federated Learning with Sketching", ICML 2020.

7. I also suggest the author perform some empirical evaluation to show the effectiveness of the proposed method.

**Summary Of The Paper:**

This paper proposes a novel federated learning framework where the client only submits a sketched gradient to the server and de-sketches the average gradient received from the server. The proposed framework can preserve the privacy of the data as well as reduce the communication cost. The paper theoretically analyzes the convergence rate and shows that the proposed method requires less communication cost than existing methods.


**Summary Of The Review:**

The paper lacks empirical results and the theoretical analysis is somewhat weak. Thus I recommend to reject the paper.

---

> ### Author Response · Authors · 2021-11-22
> **Thank you for your review and comments!**
>
> We thank the reviewer for the valuable comments and questions. We respond to them as follows:
>
> > For instance, the paper uses $a$ to denote the value of coordinate-wise embedding in section 4.1, but uses $α$ instead in section 5 and appendix.
>
> We point out that they are different notations. The notation $a$ represents a parameter in defining the coordinate-wise embedding property, whose value is $O(1)$ for many commonly used random matrices such as random Gaussian, AMS, SRHT, etc.. While the notation $\alpha$ in Section 5 corresponds to the whole term $a\cdot \frac{d}{b_{sketch}}$, which is the excessive update perturbation introduced due to randomization. The order of $\alpha$ is $O(\frac{d}{b_{sketch}})$. We notice that this confusion causes some of the following questions, especially those related to the proofs, we thank the reviewer for the careful reading and will explain this point again in the following responses.
>
> > The theorems in section 5 only require Assumption 3.1 holds, but I think they also require theorem 4.2 holds since their proofs use the property of the sketching matrices.
>
> Exactly. All the random matrices discussed in this work satisfy the coordinate-wise embedding property with $a=O(1)$. We will add this assumption to the theorems.
>
> > The inequality below Eq. (18) says E[destt(skt(h))]≤(1+α)||h||2. However, Thm 4.2 only tells us E[destt(skt(h))]≤(1+αd/bsketch)||h||2. The paper should provide some explanation on why it can remove the d/bsketch term?
>
> We point out that $a$ and $\alpha$ are different notations. In Thm 4.2, we have notation $a$, which is a constant. And in Eq. (18), we use the notation $\alpha$, which is equal to $ad/b_{sketch}$. So we did not remove any term here.
>
> > I'm also confused with the proof of Corollary 5.7 and 5.9 (Thm F.8 and Thm F.10). In the last sentence of the proof of Thm F.8 says "the optimal α  is given by O(d)." Since α is the value of the coordinate-wise embedding of the sketching matrix, how to let α be O(d)? The proof of Thm F.10 faces the same problem.
>
> Here $\alpha$ means $ad/b_{sketch}$. The value $a$ is $O(1)$ for commonly used random matrices such as random Gaussian. The sketching dimension $b_{sketch}$ is at least 1. That is why $\alpha = ad/b_{sketch}$ is at most $O(d)$ and can be as large as $O(d)$.
>
> > For Thm 5.6 and 5.8, η=1/(8(1+α)LK) is not enough to guarantee wT converges to the w∗ since the last term in the bounds won't decrease when the number of iteration increase. I think η should be related with T or ϵ.
>
> We point out that the result of Thm 5.6 and 5.8 are not saying $w^T$ converge to the $w^∗$, but only give an upper bound on their distance when $\eta$ is small enough. To ensure $w^T$ will be $\epsilon$-close to $w^∗$, we do need $\eta$ depending on $\epsilon$, as presented in Corollary 5.7 and Corollary 5.9.
>
> > It seems that the clients use different sketching matrices for different t. How to guarantee that all clients use the same sketching matrix in each round without communication?
>
> Great question! Our answer is that we do not need the same sketching matrices, or in other words, the same sketching and de-sketching process for each client. Note such independent sketching processes only favor us in both privacy concerns and convergence analysis. For the privacy part, each client has no information about the sketching of other clients, so that information leaking of one client will not impact others. For the convergence part, using the independent sketching favors us in controlling the excessive noise introduced due to randomization. And it is not hard to see that our analysis directly extends to the case of different sketching matrices by introducing a client-specific sketch/de-sketch operator $desk_t$ and $sk_t$.
>
> > The following papers also adopt sketching to reduce the communication cost of federated learning. I suggest the author discuss the difference between those papers and this work.
>
> Our work presents sketching techniques where a wide range of sketching matrices with coordinate-wise embedding properties can all apply, including commonly used random Gaussian, AMS, SRHT [1], sparse embedding [2, 3, 4, 5], etc. While both works mentioned by the reviewer only discuss the specialized Count Sketch [6, 7] matrices.
>
> [1] Ailon and Chazelle. Approximate Nearest Neighbors and the Fast Johnson-Lindenstrauss Transform. STOC 2006.
> [2] Dasgupta, Kumar and Sarlos. A Sparse Johnson--Lindenstrauss Transform. STOC 2010.
> [3] Kane and Nelson. Sparser Johnson-Lindenstrauss Transforms. JACM 2014.
> [4] Nelson and Nguyen. OSNAP: Faster numerical linear algebra algorithms via sparser subspace embeddings. FOCS 2013.
> [5] Cohen, Jayram and Nelson. Simple Analysis of the Sparse Johnson-Lindenstrauss Transform. SOSA 2018.
> [6] Charikar, Chen and Farach-Colten. Finding Frequent Items in Data Streams. ICALP 2002.
> [7] Clarkson and Woodruff. Low Rank Approximation and Regression in Input Sparsity Time. STOC 2013.

---

> > ### Comment · Reviewer_RjL4 · 2021-11-29
> > **Thank you for your reply**
> >
> > After reading your response and other reviews, I have decided to keep my score. As pointed out by most reviewers, the paper writing needs to be improved significantly. Also, the literature review part needs to be enriched.

---

### Official Review · Reviewer_ujZn · 2021-11-03

**Correctness:** 3
**Technical Novelty And Significance:** 3
**Empirical Novelty And Significance:** Not applicable
**Recommendation:** 5
**Confidence:** 4

**Main Review:**

Strengths:

1) The paper considers an important problem of addressing both communication cost and privacy concerns in the federated learning setting.

2) The paper makes several interesting technical contributions in the federated learning setting. In particular, the proposed sketching/de-sketching based learning algorithm is simple yet effective (as demonstrated by the analysis in the paper).

Weaknesses:

1) There is significant room for improvement in the presentation of the paper. The paper is filled with typos (see below) and many statements need to be paraphrased to effectively communicate the intended meaning.

2) Even though the paper introduces privacy as one of the main focus points, it briefly discusses the privacy guarantee of the proposed method in Section 4.3. The authors should at least present a formal statement/results outlining the privacy guarantee. Moreover, the significance of considering two gradient vectors that differ in the single coordinate is not immediately clear. Shouldn't the authors focus on the two gradient vectors resulting from two local datasets that differ in a single data point? Please add a more detailed discussion around this.

3) In Section 4.3, the authors justify presenting the convergence results for only a noiseless setting. However, it appears that variance of the noise added to the gradient with d. How does such a large noise affect the convergence results? It would be interesting to present the results that take the presence of added noise into account.

4) The authors have provided a pretty extensive supplementary. It would be nice if the authors can connect the supplementary with the main text with more references to the supplementary.

5) The authors discuss the issue of increased computation due to the increased number of iterations at the end of the paper. Please also consider discussing this issue earlier, e.g., in the introduction.

6) There is scope for improving the discussion of related work. The statements like "...no work addresses both challenges simultaneously as far as we [are] concern[ed]" are not clear to the reviewer. For example, there is this prior work (https://arxiv.org/abs/1805.10559) that explores the issue of communication cost and privacy.

7) Could the authors comment on their claim that they don't increase the communication cost. As far as I can see, the authors only show that the communication cost remains the same in *order-wise sense*. Please consider modifying the relevant statements accordingly. (Some empirical results that support the authors' claims will be valuable here.)

8) The paper considers a gradient descent style algorithm where all clients participate in each round. Could the author comment on how their proposed solution should be modified for the setting where only a subset of clients participates in each round.

9) Please define $\Pi$ in Definition 4.1

10) In Section 4.2, please elaborate on "(1 \pm \epsilon) guarantee".

11) Please consider paraphrasing the discussion in the second paragraph of Section 4.3 more precisely.

Some examples of typos:

1) page 1, '..the amount of data to communication between...' --> '..the amount of data to be communicated between...'
2) page 1, '...is one of the most important the key...' --> please drop either 'the most important' or 'the key'
3) page 2, 'Fl' --> 'FL'
4) page 2, '...a central server update...' --> '...a central server updates...'
5) page 2, '...three unique challenge...' -->  '...three unique challenges...'
6) page 3, '....applications in numerical linear.' --> '....applications in numerical linear algebra.' ?
7) page 4, '...we only communicates a low-dimensional sketched gradients.' --> '...we only communicate low-dimensional sketched gradients.''
8) The authors use both 'Section' and 'section'. Please consistently use one of these.
9) page 6, '...since they intuitively, they ...' --> '...since they...'

[There are similar typos in other places. Please proofread to fix those.]


**Summary Of The Paper:**

This paper proposes a sketching-based federated learning algorithm to address the communication and privacy concerns in the federated learning setting. In the proposed algorithm, each client projects its local update (gradient) to a lower dimension via a sketching matrix before communicating the update to the server. The server aggregates the sketched local updates from different clients and communicates the aggregated update to the clients. The clients then de-sketch the message received from the server to update their local model.

The paper shows that if the sketching matrix satisfies the "coordinate-wise embedding" property, then the proposed algorithm leads to provable convergence for smooth and convex objective functions.

The paper also discusses the issue of privacy and proposes to add Gaussian noise to the sketched gradients to claim differential privacy of the resulting learning algorithm.

**Summary Of The Review:**

Overall, the paper explores an interesting and timely problem. The paper presents a simple and effective scheme that has provable performance guarantees. However, there is significant room for improvement to make the contributions of the paper clear. Especially, the discussion around privacy guarantees needs to be improved. The authors also need to discuss missing prior work on communication efficiency + privacy for federated learning.

---

> ### Author Response · Authors · 2021-11-22
> **Thank you for your review and comments!**
>
> We thank the reviewer for the valuable comments and questions. We respond to them as follows:
>
> > The paper is filled with typos (see below) and many statements need to be paraphrased to effectively communicate the intended meaning.
>
> Thank you for pointing them out. We will carefully proofread to fix all the typos and grammar issues.
>
> >  the significance of considering two gradient vectors that differ in the single coordinate is not immediately clear. Shouldn't the authors focus on the two gradient vectors resulting from two local datasets that differ in a single data point?
>
> Our approach for privacy follows from Definition 1 of work [1] that discusses differential privacy upon the gradient vectors with at most 1 different attribute value.
>
> [1] Privacy via the Johnson-Lindenstrauss Transform, Krishnaram Kenthapadi, et al. 2012
>
> > However, it appears that variance of the noise added to the gradient with d. How does such a large noise affect the convergence results? It would be interesting to present the results that take the presence of added noise into account.
>
> Great question! We point out that the noise level essentially depends on $d/b_{sketch}$, which is the sketch ratio for each communication step. In the added noise case, we will have an additional term in iteration/communication complexity to characterize the impact of the constant-level added noise. For iteration complexity, this term depends linearly on this sketching ratio $d/b_{sketch}$. For communication complexity (number of bits transmitted), this term reduces back to $d/b_{sketch}\cdot b_{sketch} =d$, which matches the same order of the communication complexity of a classical local SGD.
>
> > It would be nice if the authors can connect the supplementary with the main text with more references to the supplementary.
>
> Thank you for the suggestion. We will add a detailed roadmap to clarify the structure of the supplementary material.
>
> > The authors discuss the issue of increased computation due to the increased number of iterations at the end of the paper. Please also consider discussing this issue earlier, e.g., in the introduction
>
> Thank you for the suggestion. Such computation-communication trade-off is natural and inevitable in some sense. However, for the task of Federated Learning, the communication cost is usually considered at a higher priority or a bottleneck compared to computation. We will discuss this issue in the introduction.
>
> > For example, there is this prior work (https://arxiv.org/abs/1805.10559) that explores the issue of communication cost and privacy.
>
> Thank you for pointing it out. We will add a well-rounded discussion of the related work including the ones mentioned by the reviewer.
>
> > Could the authors comment on their claim that they don't increase the communication cost.
>
> We are considering the order of the total communication cost, i.e., how it depends on the data/gradient dimension, the complexity of the objective function, etc. In this order-wise sense, the total communication cost of our approach is the same as the classical approach.
>
> > The paper considers a gradient descent style algorithm where all clients participate in each round. Could the author comment on how their proposed solution should be modified for the setting where only a subset of clients participates in each round.
>
> Good question! The essence of our approach is to apply randomized compression techniques when the clients send their message to the server, and the server de-compresses to retrieve the information with an informatic guarantee. Therefore, our approach directly extends to the case where in each iteration, only a subset of clients sends their message to the server. The only modification is changing line 5 of Algorithm 1 to a random subset of clients instead of all clients.
>
> > Please define \Pi in Definition 4.1
>
> $\Pi$ is the distribution of the random matrix R discussed in Definition 4.1.
>
> > In Section 4.2, please elaborate on "(1 \pm \epsilon) guarantee".
>
> The classical sketching is able to choose an appropriate sketching dimension, such that the solution to the sketched problem has the guarantee of being ($1 \pm \epsilon$) optimal to the original problem, where $\epsilon$ depends on the sketching dimension as well as the type of the sketching matrix.
>
> > Please consider paraphrasing the discussion in the second paragraph of Section 4.3 more precisely.
>
> Thank you for the suggestion. We will make the discussion of 4.3 more clear and precise.

---

> > ### Comment · Reviewer_ujZn · 2021-11-29
> > **Keeping my original score**
> >
> > Thank you for your response. After going through your response and other reviews, I have decided to keep my original scores. Besides improving the presentation of the paper, please consider highlighting the differences with prior work pointed out by all the reviewers.

---

### Decision · Program_Chairs · 2022-01-20

**Decision:**

Reject

**Comment:**

The paper studies federated learning with various sketching techniques used for communication.

The main concerns from the reviewers are:

1) the presentation can be improved;

2) the novelty and related work section is not satisfactory since there have been papers on sketched federated learning;

3) there is no numerical study to validate the efficacy of the method.

I suggest the authors to take into consideration the feedback from the reviewers in the revision of the paper.